# Probing metazoan polyphosphate biology using *Drosophila* reveals novel and conserved polyP functions

**Sunayana Sarkar[1], Harsha Sharma[1], SK Yasir Hosen[1], Jayashree S Ladke[2,3], Sandra Moser[4,5], Deepa Balasubramanian[1], Sreejith Raran-Kurussi[1], Henning J Jessen[4,5], Rashna Bhandari[2]\*, Manish Jaiswal[1]\***

[1]Tata Institute of Fundamental Research, Hyderabad, India; [2]Centre for DNA Fingerprinting and Diagnostics, Hyderabad, India; [3]Graduate Studies, Regional Centre for Biotechnology, Faridabad, India; [4]Institute of Organic Chemistry, Albert-Ludwigs-Universität Freiburg, Freiburg im Breisgau, Germany; [5]CIBSS – Centre for Integrative Biological Signalling Studies, Albert-Ludwigs-Universität Freiburg, Freiburg im Breisgau, Germany

**\*For correspondence:**
rashna@cdfd.org.in (RB);
manish@tifrh.res.in (MJ)

**Competing interest:** The authors declare that no competing interests exist.

## eLife Assessment

Studying the biological roles of polyphosphates in metazoans has been a longstanding challenge to the field given that the polyP synthase has yet to be discovered in metazoans. This **important** study capitalizes on the sophisticated genetics available in the Drosophila system and uses a combination of methodologies to start to tease apart how polyphosphate participates in Drosophila development and in the clotting of Drosophila hemolymph. The data validating one of these tools (cyto-FLYX) are **solid** and well-documented and they will open up a field of research into the functional roles of polyP in a metazoan model. The other tools for tissue specific knockdown of polyP (Mito-FLYX, ER-FLYX, and Nuc-FLYX) have not yet been validated but will be invaluable to the field when they are.

**Abstract** Polyphosphate (polyP) exists in all life forms; however, its biological functions in metazoans are understudied. Here, we explored *Drosophila,* to our knowledge, as the first genetic model to explore polyP biology in metazoans. We established biochemical and in situ methods to detect, quantify, and visualise polyP in *Drosophila*. We then engineered a FLYX system to deplete polyP in subcellular compartments in a tissue-specific manner. Using these tools, we demonstrated a spatio-temporal and subcellular compartment-specific regulation of polyP levels in various developmental stages and tissue types. We discovered that polyP is crucial for *Drosophila* hemolymph clotting and proper developmental timing, consistent with an evolutionarily conserved role as exogenous polyP also accelerates mammalian blood clotting. Furthermore, the transcriptomics analysis of polyP-depleted larvae demonstrates the impact of polyP on several cellular processes, including translation. These observations underscore the utility of the toolkit we developed to discover previously unknown polyP functions in metazoans.

## Introduction

Inorganic polyphosphate (polyP), a linear polymer of variable chain lengths of orthophosphate (Pi) moieties linked via phosphoanhydride bonds, was first observed in the bacteria *Spirillum volutans* and was termed as volutin granules (*Wiame, 1947*; *Achbergerová and Nahálka, 2011*). Arthur Kornberg's

works showed that polyP is present in archaea, eubacteria, and mammals (*Kornberg et al., 1999*; *Kornberg, 1995*). Our current understanding of the biological functions and the molecular mechanisms of polyP predominantly stems from studies in prokaryotes and a few single-cell eukaryotes (e.g. *Saccharomyces cerevisiae, Schizosaccharomyces pombe,* and *Dictyostelium sp.*). In bacteria, polyP is shown to serve as a phosphate reservoir, aid in biofilm formation, and help in transposon silencing (*Cremers et al., 2016*; *Wang et al., 2020*; *Beaufay et al., 2021*). In algae, polyP acts as an antioxidant, metal chelator, and buffer against alkaline stresses (*Kornberg et al., 1999*; *Dunn et al., 1994*; *Archibald and Fridovich, 1982*; *Pick and Weiss, 1991*). However, in metazoans, the biological functions of polyP have been underexplored, and thus, it used to be referred to as a molecular fossil.

Ex vivo and in vitro works in mammals have suggested the implications of polyP in blood coagulation, mitochondrial function, bone mineralisation, and neuronal activity (*Smith et al., 2006*; *Montilla et al., 2012*; *Ghosh et al., 2013*; *Smith et al., 2010*; *Seidlmayer et al., 2012*; *Holmström et al., 2013*; *Wang et al., 2019*; *Omelon et al., 2009*). Moreover, recent works have discovered that several mammalian proteins, known to be involved in cellular processes, such as transcription and translation, can bind to polyP. Binding to polyP may alter the localisation, structure, or function of proteins, thereby affecting several biological processes in metazoans (*Azevedo et al., 2018*; *Azevedo et al., 2015*; *Neville et al., 2023*). Recent studies have also implicated polyP in human diseases, such as the pathogenesis of amyotrophic lateral sclerosis (ALS) (*Arredondo et al., 2022*). Furthermore, a clinical trial has found beneficial effects of polyP in treating ulcerative colitis (*Fujiya et al., 2020*). Furthermore, feeding polyP to aβ-expressing *Caenorhabditis elegans* (Alzheimer's model) led to rescuing paralytic behaviour (*Cremers et al., 2016*). Thus, it is emerging that polyP is not a relic; it affects several physiological processes in metazoans, including human diseases.

Efficient tools for quantification and genetic manipulation of polyP in prokaryotic models and yeast have been crucial in exploring polyP biology. Discovery of PolyP kinases (PPK1 in bacteria, VTC4 in yeasts) and Polyphosphatases (PPN2 in bacteria, ScPPX in yeasts) - enzymes to synthesise and degrade polyP in prokaryotes and yeasts allowed phenotypic studies in these organisms (*Desfougères et al., 2020*; *Zhu et al., 2005*; *Wei et al., 2015*; *Rangarajan et al., 2006*; *Bolesch and Keasling, 2000*; *Ugochukwu et al., 2007*; *Alvarado et al., 2006*). In contrast, our knowledge of polyP and its roles remains limited in multicellular eukaryotes. Although recent work has reported that the FoF1 complex of ATPase in the mitochondria is required for polyP synthesis, PolyP kinases, and polyphosphatases in multicellular organisms remain unknown (*Baev et al., 2020*). Purified human Prune is shown to have a short-chain exopolyphosphatase activity in vitro (*Tammenkoski et al., 2008*). Recently, biochemical experiments led to the discovery of endopolyphosphatase NUDT3, an enzyme known as a dinucleoside phosphatase (*Samper-Martín et al., 2021*). The function of these proteins in regulating polyP has yet to be tested in vivo in a genetically tractable metazoan model.

A significant gap in studying polyP in multicellular eukaryotes is the lack of a genetic model system that allows sensitive methods for visualising, quantifying, and manipulating polyP. *Drosophila melanogaster* provides an ideal platform to close this gap: it combines powerful genetics with short generation time and well-characterized development, enabling systematic interrogation of polyP dynamics in vivo. However, methods for sensitive quantification, visualisation, and genetic perturbation of polyP have not been established in flies or any other metazoan model. Here, we report a toolkit to quantify, visualise, and genetically manipulate polyP levels in *Drosophila*. Using this toolkit, we show that polyP levels undergo temporally regulated fluctuations that can influence organismal development and physiology through possible evolutionarily conserved mechanisms. Overall, this work establishes *Drosophila* as a valuable in vivo model to dissect the functions of polyP in metazoans.

## Results
### Development of methods for the extraction and quantification of polyP from flies

The polyP levels have been shown to vary among species. Bacteria like *E. coli* maintain as low as 100 uM polyP (Pi terms), which surges to nearly 50 mM in stress conditions; yeasts have been reported to have a polyP content of ~100 mM (Pi terms), whereas in mammalian cells, polyP levels vary in low micromolar concentrations (~10–90 uM polyP in Pi terms) (*Kornberg et al., 1999*). Since metazoans have lower polyP content, to use *Drosophila* as a model system, it was essential to re-standardise a

method sensitive enough to extract and quantify polyP from fly samples. Various biochemical strategies for extracting polyP from biological samples have been discussed, each with its own limitations (**Bru et al., 2016**). The acid-based extraction method fails to preserve the entire chain length of polyP, as polyP is unstable in strongly acidic conditions. In contrast, column purification can only retain polyP of longer chain lengths (>60–80). PolyP extraction using saturated phenol (in citrate buffer pH 4.1) and chloroform, followed by ethanol precipitation, can efficiently pellet down polyP. However, this method also co-purifies RNA, which may interfere with polyP quantification. Thus, to extract polyP from *Drosophila* samples, we standardised the citrate-saturated phenol-chloroform-based method followed by RNase treatment (**Figure 1A**, **Figure 1—figure supplement 1A**, Materials and methods).

To quantify polyP levels, several techniques based on Nuclear Magnetic Resonance (NMR), chromatography, radioactive isotope labelling, and enzymatic digestion have been used (**Christ et al., 2020**). We used the enzymatic detection method to quantify polyP extracted from fly tissues, as it has higher specificity and sensitivity (**Christ and Blank, 2018**; **van Veldhoven and Mannaerts, 1987**). This method involved the digestion of polyP by recombinant *S. cerevisiae* exopolyphosphatase 1 (*Sc*PpX1), followed by colorimetric measurement of the released Pi by malachite green. Malachite green and ammonium molybdate, when mixed in a 3:1 ratio, form a metachromatic brown-coloured adduct that turns green upon conjugation with monophosphate (**Petitou et al., 1978**; **Hohenwallner and Wimmer, 1973**) and can be quantified based on absorbance at 650 nm. Therefore, absorbance measurements from *Sc*PpX1 digested polyP can be interpolated to quantify the Pi released from extracted polyP samples. Thus, the intensity of the green colour is a measure of the free Pi, which in turn is a measure of the polyP levels. This assay allows the measurement of polyP content indirectly in terms of the liberated Pi (**Figure 1—figure supplement 1B**, Materials and methods).

To standardise the polyP quantification method, we extracted polyP from fly larvae, and the extracted polyP was incubated with purified recombinant *Sc*PpX1 for 18 hr at 37°C to digest polyP completely. The other half was left untreated for 18 hr at 37°C. Malachite green was added to both samples, and the background from any preexisting Pi in the untreated sample was subtracted to quantify the Pi content. For calibration of the assay, increasing concentrations of $KH_2PO_4$ were used to draw a standard curve (**van Veldhoven and Mannaerts, 1987**; **Petitou et al., 1978**; **Hohenwallner and Wimmer, 1973**). The amount of recovered polyP was proportional to the number of larvae used for polyP extraction (**Figure 1B**). The polyP content in each third instar larval was found to be 419.30±36.83 picomoles in Pi terms (**Table 1**), and the normalised polyP level is 46.8±3.95 picomoles polyP (Pi terms) per milligram protein from the third instar larval lysate (**Table 1**).

## PolyP levels are developmentally regulated

Given the various possible roles of polyP, we sought to test if polyP levels are developmentally regulated in *Drosophila*. First, we quantified polyP at every 2 hr interval during the embryonic stages of development. As shown in **Figure 1C** and **Table 2**, we observed no significant change in polyP levels across various embryonic stages. We then compared polyP content at different stages of the *Drosophila* life cycle - embryos, larval stages (first instar, second instar, feeding third instar, non-feeding third instar, pupal stages prepupae and one-day-old pupae), pharate adult, and three-days-old adults. The polyP content remains unchanged from embryos to the third instar feeding larval stages, followed by an increase in the third instar non-feeding larval stage. The polyP content reduces during metamorphosis and drops significantly in the pharate, the stage when the adults are ready to eclose from pupa, followed by a subtle increase in three-day-old flies (**Figure 1D**, **Table 3**). These observations show a temporal regulation of polyP during development, and the increased polyP levels at the late third instar larval stages could constitute a phosphate reservoir for metamorphosis during pupal stages.

## In-situ labelling of polyP in *Drosophila* tissues revealed spatiotemporal polyP dynamics

To investigate the subcellular localisation of polyP across different tissue types, we developed an in situ label of polyP using the <u>P</u>oly<u>P</u> <u>B</u>inding <u>D</u>omain (PPBD) of the *E. coli* exopolyphosphatase PPX. PPBD was initially shown to specifically bind polyP in vitro with a high affinity of ~45 μM for longer polyP chains (35 mer and above), and it can detect polyP localized in budding yeast vacuoles, mammalian mast cell granules, and *C. elegans* (**Saito et al., 2005**; **Moreno-Sanchez et al., 2012**;

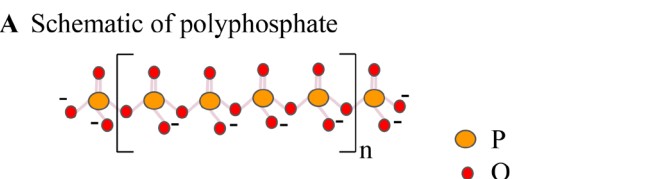

**A** Schematic of polyphosphate

**B** Quantitative estimation of polyP from *Drosophila* larvae

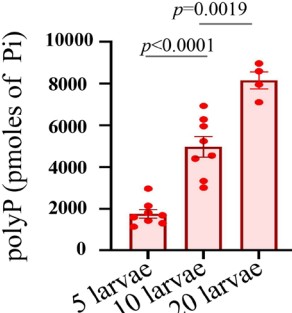

**C** Estimation of polyP from various embryonic stages

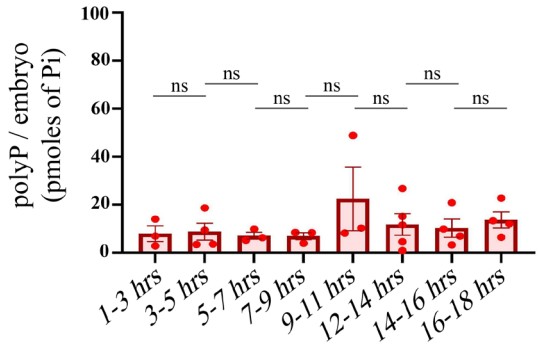

**D** Estimation of polyP from various life cycle stages

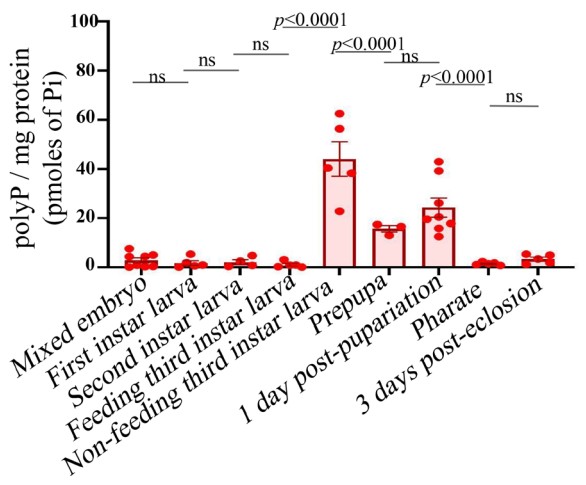

**Figure 1.** Polyphosphate extraction and quantification from flies. (**A**) Linear polyphosphate. (**B**) Polyphosphate (PolyP) quantification from sets of 5, 10, 20 third instar larvae (in Pi terms), N=5 in each set. Statistics: Student's *t*-test *p*>0.001, error bar - s.e.m. *Drosophila* strain used - *CantonS*. (**C**) Quantification of polyphosphate across embryonic stages. n=75 in each time point, N=3. (**D**) Quantification of polyP across a fly life cycle: embryos (n=75,

*Figure 1 continued on next page*

*Figure 1 continued*

N=8), first instar larvae (n=25, N=5), second instar larvae (n=25, N=5), feeding third instar larvae (n=10, N=5), non-feeding third instar larvae (n=10, N=5), prepupae (n=20, N=3), one-days-post-pupariation pupae (n=20, N=8), pharate (n=10, N=5), three-days-post-eclosion adult (n=10, N=5). Statistics: Student's *t*-test with *p*>0.001, error bar - s.e.m. *Drosophila* strain used - *CantonS*.

The online version of this article includes the following figure supplement(s) for figure 1:

**Figure supplement 1.** Schematic of polyP extraction and estimation.

---

*Moreno-Sanchez et al., 2012*; *Negreiros et al., 2018*; *Quarles et al., 2024*). Thus, in this study, we used recombinant GST-PPBD protein to label polyP in fly tissues (*Figure 2A*, Materials and methods).

To standardise the labelling method, we determined a tissue fixation method and tested methanol, 4% paraformaldehyde, and Bouin's fixatives and found that 4% paraformaldehyde works better among the three for labeling polyP in fly tissues (*Figure 2—figure supplement 1A*). To test the specificity of GST-PPBD, we used a mutant PPBD (GST-PPBD^Mut) and GST proteins. To create GST-PPBD^Mut we replaced eight basic amino acid residues with Alanine (*Figure 2B*, *Supplementary file 1* for sequence). These amino acids were selected based on structural studies of *E. coli* PPX that contains PPBD domain (*Rangarajan et al., 2006*; *Alvarado et al., 2006*). The predicted structure of the PPBD^Mut aligned with PPBD and had a r.m.s.d. value of 0.024 suggesting no major change in structure of the protein (*Figure 2B*). We synthesised diamine-linked polydisperse PolyP$_{100}$ and labelled both the ends with Fluorescein Isothiocyanate (FITC) to yield Bis-FITC-labeled PolyP$_{100}$ (PolyP$_{100}$-2X-FITC) following a synthetic procedure described by *Fernandes-Cunha et al., 2018* (Materials and methods). Using Microscale Thermophoresis (MST), we found that GST-PPBD binds to PolyP$_{100}$-2X-FITC whereas GST-PPBD^Mut and GST do not show binding to PolyP$_{100}$-2X-FITC (*Figure 2C*, Materials and methods). We then performed immunofluorescence experiments with GST, GST-PPBD, and GST-PPBD^Mut on hemocytes and found that, unlike GST-PPBD, GST-PPBD^Mut, or GST protein does not show any specific staining pattern (*Figure 2D*). Thus, GST-PPBD^Mut and GST protein can serve as checkpoints for the specificity of GST-PPBD based labeling of polyP. To further test the specificity, we treated fixed and permeabilised hemocytes with *Sc*PpX1 and heat-inactivated *Sc*PpX1 for 2 hr at 37°C and found that both the staining intensity and the number of PPBD punctae were reduced in hemocytes treated with active *Sc*PpX1, as compared to untreated control and heat-inactivated *Sc*PpX1 (*Figure 2E–G*), method adopted from *Quarles et al., 2024*.

We systematically analysed GST-PPBD labelling in several fly tissues and observed distinct patterns in different tissue types (*Figure 3* and *Figure 3—figure supplement 1*). We observed GST-PPBD labelling in cytoplasmic puncta, nuclear compartments, or both in a tissue-specific manner (*Figure 3* and *Figure 3—figure supplement 1*). In cells with larger nuclei, *e.g.*, in salivary glands, we observed intense GST-PPBD labelling in subnuclear compartments devoid of DAPI staining (*Figure 3A ii, iv*). We suspected these compartments to be nucleoli. To test this, we co-stained GST-PPBD with an antibody against fibrillarin, a nucleolar protein, and found colocalization of these proteins, suggesting that polyP is enriched in the nucleolus (*Figure 3B–C*). This is consistent with recent reports, which found that polyP binds to nucleolar proteins (*Borghi et al., 2024*; *Jimenez-Nuñez et al., 2012*). Furthermore, in wing and eye larval imaginal discs, we found punctate cytoplasmic and nucleolar labelling of GST-PPBD. However, leg and haltere imaginal discs showed no difference in GST-PPBD staining compared to the GST control. In contrast, the larval crop and muscle showed nucleolar GST-PPBD with an intense cytoplasmic background (*Figure 3—figure supplement 1*).

Interestingly, GST-PPBD labelling revealed spatial and temporal dynamics of polyP localisation during oogenesis. We observed intense GST-PPBD labelling in the nucleus and cytoplasm of the nurse and follicle cells in the early stages of ovary development (*Figure 3D*). However, there is a gradual reduction in the intensity of GST-PPBD labelling from stage S2 to S9 ovaries, indicating a

**Table 1.** Absolute quantification of Polyphosphate (polyP) levels.

| | PolyP quantification |
|---|---|
| Normalised to protein content | 46.80±3.95 picomoles in Pi terms/mg protein |
| Normalised to number of larvae | 419.3±36.83 picomoles in Pi terms/larva |

**Table 2.** Polyphosphate (PolyP) quantification from embryo (*Figure 1C*).

| Embryo stages (hours) | polyP (pmoles of Pi/embryo) |
| --- | --- |
| 1–3 hr | 7.961±3.27 |
| 3–5 hr | 8.825±3.57 |
| 5–7 hr | 7.135±1.39 |
| 7–9 hr | 6.930±1.48 |
| 9–11 hr | 22.47±13.22 |
| 12–14 hr | 14.57±4.49 |
| 14–16 hr | 10.26±3.78 |
| 16–18 hr | 13.67±3.38 |

gradual decrease in polyP levels, indicating the potential stage-specific physiological role of polyP (*Figure 3D*). Earlier reports have also demonstrated polyP localisation in the zona pellucida of mice ovaries and in cockroach oocytes, indicating a possible role of polyP during oogenesis (*Nakamura et al., 2018*; *Motta et al., 2009*; *Gomes et al., 2008*). In hemocytes (hemolymph cells, equivalent to mammalian blood cells), we found an intense GST-PPBD labelling exhibiting polyP localisation in large granular structures close to the periphery (*Figure 3E–H*). The hemocytes were identified by the expression of GFP reporters, driven by cell-specific promoters of *hemolectin* (*hml-GAL4 >UAS* GFP) in plasmatocytes and *lozenge* (*lz-GAL4 >UAS-mCD8::GFP*) in crystal cells. Overall, to our knowledge, this is the first comprehensive tissue staining report exhibiting the spatiotemporal dynamics of polyP within a multicellular organism. Taken together, these data also indicate tissue-specific regulation and biological function of polyP in metazoans.

## A heterologous system to deplete polyP in flies: FLYX- Fly expressing *Sc*PpX1

Since metazoan polyP kinases and polyphosphatases are still elusive, direct genetic manipulation of polyP levels can not be done to probe their function in flies. Heterologous expression of budding yeast exopolyphosphatase enzyme (*Sc*PpX1) can deplete polyP in mammalian cells (*Wang et al., 2003*). Therefore, to investigate the biological function of polyP, we developed transgenic fly lines that allow tissue-specific and subcellular depletion of polyP by heterologous expression of *Sc*PpX1. These flies that will express *Sc*PpX1 constitute the 'FLYX' system.

To develop the FLYX system, we codon optimised and synthesised N-terminal HA-tagged *Sc*PpX1 (HA-*Sc*PpX1) (*Supplementary file 1*). For the tissue-specific expression of HA-*Sc*PpX1 in the FLYX system, we used the UAS-GAL4 system (*Duffy, 2002*; *Brand and Perrimon, 1993*). The transcription activator GAL4 is expressed under a tissue-specific promoter in this system. When GAL4 binds to the

**Table 3.** Polyphosphate (PolyP) quantification across fly life cycle (*Figure 1D*).

| Developmental stages | polyP (pmoles of Pi/mg protein) |
| --- | --- |
| Embryo | 2.808±1.0 |
| First instar | 1.714±0.94 |
| Second instar | 2.038±1.0 |
| Feeding third instar | 1.002±0.53 |
| Non-feeding third instar (wandering) | 44.06±7.04 |
| Prepupa | 15.63±1.34 |
| One day post-pupariation | 16.88±3.93 |
| Pharate | 1.398±0.24 |
| Three days post-eclosion | 2.445±0.78 |

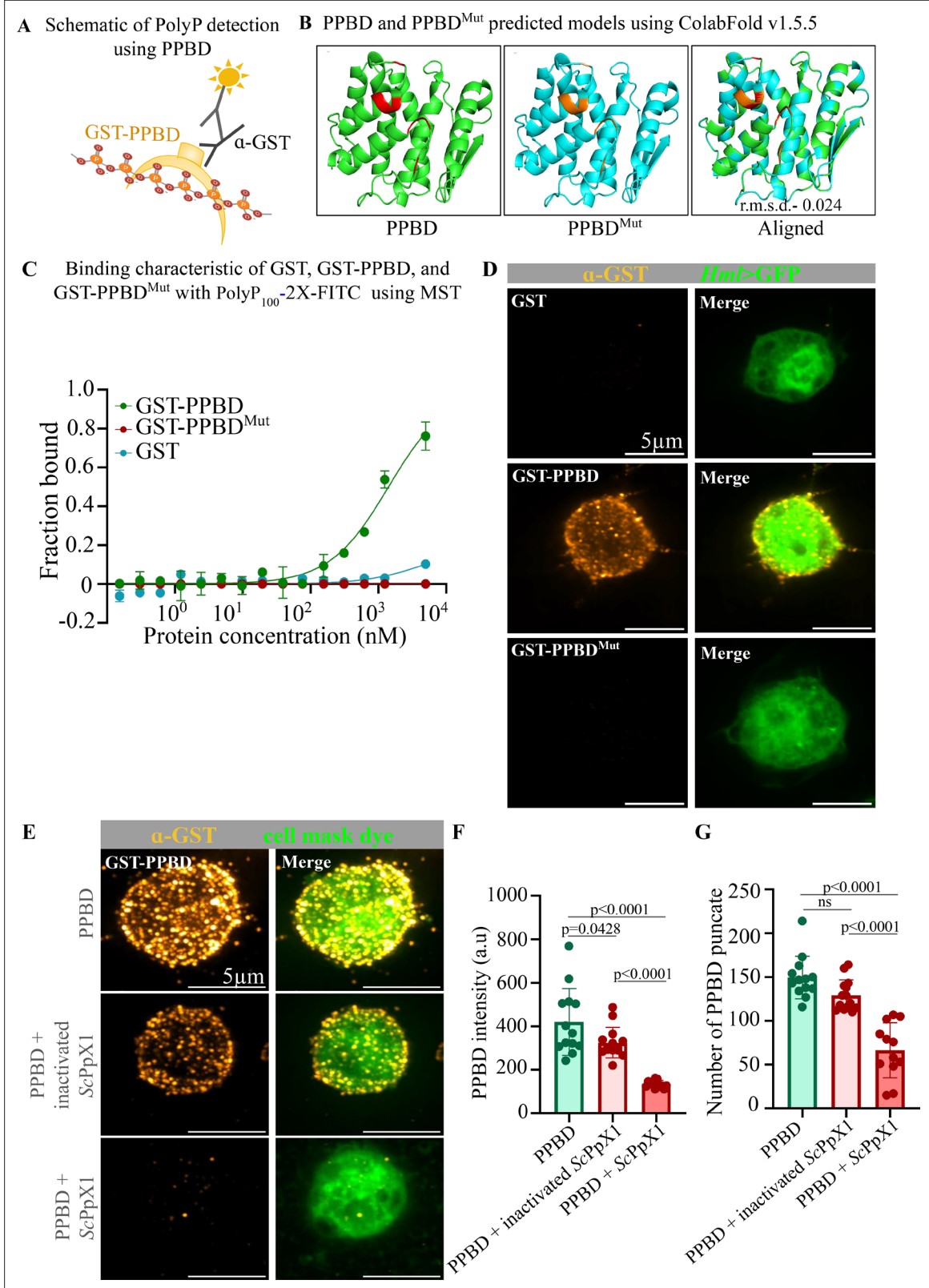

**Figure 2.** Validation of PolyP Binding Domain (PPBD) as a probe for polyphosphates in fly tissues. (**A**) Schematic of Polyphosphate (PolyP) detection using PPBD fused at the N-terminus with a GST tag. (**B**) PPBD and PPBD$^{Mut}$ predicted structural models using ColabFold v1.5.5: AlphaFold2 using MMseqs2 (free online). The amino residues marked in red are changed to Alanine (marked in orange). Overall alignment shows a r.m.s.d. Value of 0.024. (**C**) Binding characteristic of GST, GST-PPBD, and GST-PPBD$^{Mut}$ with polyP$_{100}$-2X-FITC using MST. Graph showing GST-PPBD fraction bound to

*Figure 2 continued on next page*

*Figure 2 continued*

polyP, whereas the GST-PPBD^Mut and GST does not bind to polyP$_{100}$-2X-FITC. (**D**) Staining of hemocytes with GST, GST-PPBD, and GST-PPBD^Mut. Hml-GAL4>UAS GFP reporter was used to identify plasmocytes. Negative control cells incubated with GST and GST-PPBD Mutant proteins separately and stained with anti-GST antibody (GST; orange hot). For polyP staining, cells were incubated with GST-PPBD protein and stained with anti-GST antibody (GST-PPBD-polyphosphate; orange hot). Scale bar - 5 µm. (**E**) Staining of hemocytes with GST-PPBD in fixed and permeabilised hemocytes with 2 hr pretreatment with buffer (control) and heat inactivated *Sc*PpX1, and active *Sc*PpX1 proteins. GST-PPBD proteins were stained with anti-GST antibody (GST-PPBD-polyphosphate; orange hot), and the cell is marked by Cell Mask Membrane staining dye (green). Scale bar - 5 µm. (**F**) Graph showing reduced anti-GST staining intensity in hemocytes treated with *Sc*PpX1. (**G**) Graph showing reduced PPBD punctae intensity in hemocytes treated with *Sc*PpX1. Statistics: One-way Annova with *p*>0.001, error bar - s.d. *Drosophila* strain used - *CantonS*.

The online version of this article includes the following figure supplement(s) for figure 2:

**Figure supplement 1.** GST::PolyP Binding Domain (PPBD) staining in fly tissues.

upstream activator sequence (UAS) preceding the cDNA sequence of interest, the protein is expressed in the specific tissue**,** thereby providing a handle for spatial regulation of polyP (*Figure 4A–B*). We cloned codon-optimised HA-*ScPpX1* into fly transgenesis vector pUAST-AttB and pUASp, which allows site-specific integration of a construct in the fly genome via φC31 Integrase system (*Groth et al., 2004*). The construct (pUAST-HA-*ScPpX1*-AttB) was injected into embryos for its integration in the AttP40 docking site in the second chromosome, and we recovered transgenic FLYX lines. We refer to these lines as Cyto-FLYX. We then validated the expression and localisation of HA-*Sc*PpX1 (driven by *da-GAL4*) using anti-HA antibody staining and found *Sc*PpX1 localisation throughout the cytoplasm (*Figure 4C and E*). To test the polyP levels in FLYX, we used a *tubulin*-GAL4 driver, which allows ubiquitous expression of *HA-Sc*PpX1. We observed significantly decreased polyP content in Cyto-FLYX larvae compared with the control, suggesting that HA-*Sc*PpX1 can deplete polyP in flies (*Figure 4D*).

To expand the *Drosophila* polyP depletion toolkit library, we engineered HA-*ScPpX1* constructs to facilitate subcellular compartment-specific depletion of polyP. We created FLYX lines that can target *Sc*PpX1 to the nucleus (Nuc-FLYX), endoplasmic reticulum (ER-FLYX), and the mitochondria (Mito-FLYX) (*Figure 4C*). We tested the localisation of this FLYX in larval muscles using anti-HA antibody staining. We found that Nuc-FLYX intensely localises in the nuclear region devoid of DAPI (DNA) signal, suggesting probable nucleolar localisation; ER-FLYX co-localized with Calnexin, an ER protein; and Mito-FLYX co-localized with ComplexV subunit A, a mitochondrial protein (*Figure 4E*, Materials and methods for sequence). This FLYX library provides a handle for spatial polyP depletion in an organism to explore various physiological functions.

## PolyP is crucial for hemolymph clotting

Since hemocytes contribute to hemolymph clotting and ex vivo studies in humans have shown exogenous addition of polyP into blood plasma accelerates blood clotting, we reasoned that polyP may accelerate hemolymph clotting in flies (*Smith et al., 2006*; *Morrissey and Smith, 2015*; *Baker et al., 2019*; *Morrissey et al., 2012*). To assess and quantify hemolymph clotting characteristics, we established a 'hemolymph drop assay' (*Figure 5—figure supplement 2*, Materials and methods). The clotting hemolymph drop in this assay displays characteristic inward fibre-like structures perpendicular to the edge of the drop. These fibres are significantly reduced in length and number in the *hemolectin* (*hml*), a mutant known to exhibit hemolymph clotting defects (*Figure 5—figure supplement 1A–E*; *Scherfer et al., 2004*). Therefore, we used fibre number and characteristics to assess hemolymph clotting phenotypes. To test the effect of polyP on hemolymph clotting, we compared the clot characteristics of hemolymph mixed with water (blank), Pi (1.6 nmol Pi, negative control) or polyP of different average chain lengths (1.6 nmol polyP in Pi terms, equivalent to the total polyP content from two larvae). Upon exogenous addition of polyP, we found increased clot fibre numbers along the edge of the hemolymph drop (*Figure 5—figure supplement 1F–J*). Similar results were also observed when we reduced the concentration by half (125 µM polyP (Pi terms) as has been used in human clotting studies) (*Figure 5—figure supplement 1F–I*; *Smith et al., 2006*; *Smith et al., 2010*). We found that the increase in the clot number density is dependent on the polyP chain length; polyP$_{14}$ did not increase the fibre numbers and length, while PolyP$_{65}$ and PolyP$_{130}$ led to a subtle and significant increase compared to blank and negative control samples (*Figure 5—figure supplement 1K–V*).

The observations that polyP can promote hemolymph clotting ex vivo and polyP is localised to large granular structures in hemocytes prompted us to test whether polyP is crucial for hemolymph

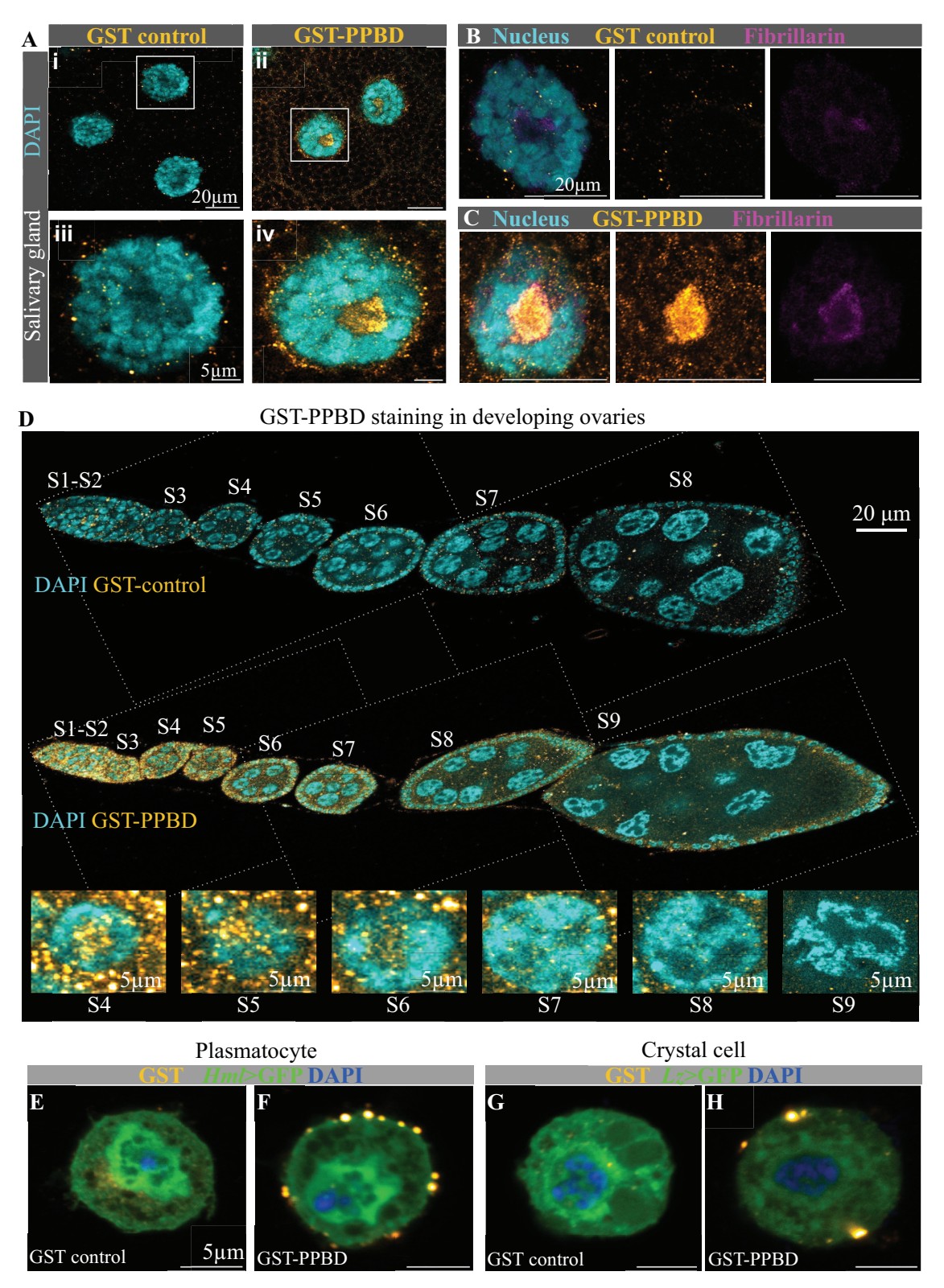

**Figure 3.** Polyphosphate staining of *Drosophila* tissues with PolyP Binding Domain (PPBD). (**A–C**) Staining of larval salivary glands with DAPI (nucleus; cyan) and anti-GST antibody (GST; orange hot). (**A i, iii**, and **B**) Samples incubated with GST (negative control). (**A ii, iv**, and **C**) Samples incubated with GST-PPBD. (**B–C**) Salivary gland stained with DAPI (cyan) anti-fibrillarin antibody (magenta) and anti-GST antibody (orange hot) to detect polyphosphate colocalisation within the nucleolus. Scale bar for (**A i, ii, B-C**)- 20 µm, scale bar for (**A iii, iv**) - 5 µm. *Drosophila* strain used - *CantonS*. (**D**) Staining of fly

*Figure 3 continued on next page*

*Figure 3 continued*

ovaries. Negative control: Incubated with GST and stained with DAPI (nucleus; cyan) and anti-GST antibody (GST; orange hot). Stages covered - S1-S8. Incubated with GST-PPBD and stained with DAPI (nucleus; cyan) and anti-GST antibody (GST-PPBD-polyphosphate; orange hot). Stages covered - S1-S9. Yellow arrow- follicle cells, Orange arrow- nurse cells. Scale bar - 20 μm. Insets reveal the nucleus and part of the cytoplasm across stages S4-S9, showing reduced polyphosphate signals. Scale bar - 5 μm. *Drosophila* strain used - *CantonS*. (**E–H**) GST-PPBD staining of hemocytes. *Hml-GAL4>UAS* GFP reporter was used to identify plasmocytes (**E–F**). *Lz-GAL4>UAS mCD8::GFP* reporter was used to identify crystal cells (**G–H**). Negative control (**E** and **G**) - cells incubated with GST protein and stained with DAPI (nucleus; blue) and anti-GST antibody (GST; orange hot). For polyP staining (**F** and **H**), cells were incubated with GST-PPBD protein and stained with DAPI (nucleus; cyan) and anti-GST antibody (GST-PPBD-polyphosphate; orange hot). Scale bar of (**A–D**) - 5 μm.

The online version of this article includes the following figure supplement(s) for figure 3:

**Figure supplement 1.** Polyphosphate staining of several *Drosophila* tissues with PolyP Binding Domain (PPBD).

clotting in vivo. Thus, we tested the effect of polyP depletion by ubiquitous expression of *Sc*PpX1 on hemolymph clotting. We observed significantly fewer clot fibre numbers, branching points, and clot fibre length in the Cyto-FLYX larvae (*tubulin* >Cyto FLYX) than the control (***Figure 5A–D***). These observations suggest that polyP is crucial for efficient hemolymph clotting.

We then sought to identify the cell type that contributes polyP for hemolymph clotting. Therefore, we decided to deplete polyP in a cell-specific manner using the FLYX system under three different drivers. We expressed Cyto-FLYX in fat bodies and all hemocytes using *cg*-GAL4 (*cg* >Cyto FLYX) and observed reduced fibre length, clot fibre numbers, and branching (***Figure 5E–H***). We then used *hml*-GAL4 (*hml* >Cyto FLYX), which mainly expresses in the plasmatocytes, and observed significantly reduced clot fibre number and branching densities. The clot fibre length, however, was comparable to the control (***Figure 5I–L***). We did not observe any significant change in clot fibre numbers and branching when we expressed Cyto-FLYX only in crystal cells using *lz*-GAL4 (*lz* >Cyto FLYX). The lengths of the clot fibres were also comparable to the control (***Figure 5M–P***). These data suggest that polyP in hemocytes is necessary for efficient hemolymph clotting. Next, we tested the effect of exogenous addition of polyP in the clotting of hemolymph from *tubulin* >Cyto FLYX flies. We divided hemolymph from *tubulin* >Cyto FLYX flies into two parts and incubated one half with Pi (negative control) while the other half was incubated with polyP$_{65}$. On polyP$_{65}$ addition, we observed no significant rescue of the relative clot fibre numbers in *tubulin* >Cyto FLYX compared to the control (***Figure 5Q-S***). This suggests that intracellular polyP in the hemocytes might have a physiological role in the clot protein secretion. Taken together, the similarities in ex vivo clotting studies in flies and humans and the in vivo studies in flies suggest that the cellular and molecular function of polyP in hemolymph/blood clotting are conserved between insects and mammals.

## PolyP affects developmental timing

Since we observed differential regulation of polyP during development and in various tissues, we sought to systematically explore phenotypic expression upon ubiquitous depletion of polyP (*tubulin* >Cyto FLYX). We found no significant differences in the weight or size of Cyto-FLYX vs. control larvae, pupae, or adults (***Figure 6—figure supplement 1A–D***). Since the pUAST vector is not efficient in driving expression in the germ line cells, we cloned Cyto-FLYX in the pUASP vector and generated a new transgenic line to deplete polyP in the germ line cells. However, we found no significant change in the fecundity upon germ line expression of Cyto-FLYX using *Nos*-GAL4 driver (***Figure 6—figure supplement 1E–F***). Furthermore, since we had observed polyP levels increase just before pupariation and decrease during pupal stages, we sought to test the impact of polyP depletion on the metamorphosis timing. We tested the total time taken from prepupae stages to the eclosion of adults when Cyto-FLYX is expressed ubiquitously (*tubulin* >Cyto FLYX, ***Figure 6A–B***). Typically, metamorphosis takes ~ 5 days, we found *tubulin* >Cyto FLYX required ~14 hr less for the eclosion of 50% pupae as compared to control (119.2±3.863 hr for *tubulin*-Cyto-FLYX and 134.5±4.877 hr for control). This data uncovers yet another impact of polyP on metazoal biology.

## PolyP depletion affects a wide range of biological processes

Nutrient conditions and signalling pathways, such as Ecdysone, insulin, and TOR are known to impact metamorphosis timing in flies. Third instar larvae undergoes multiple pulses of ecdysone signalling mediated through, among others, *ecd* (promotes metamorphosis), *Eip74EF* (ecdysteroid target gene

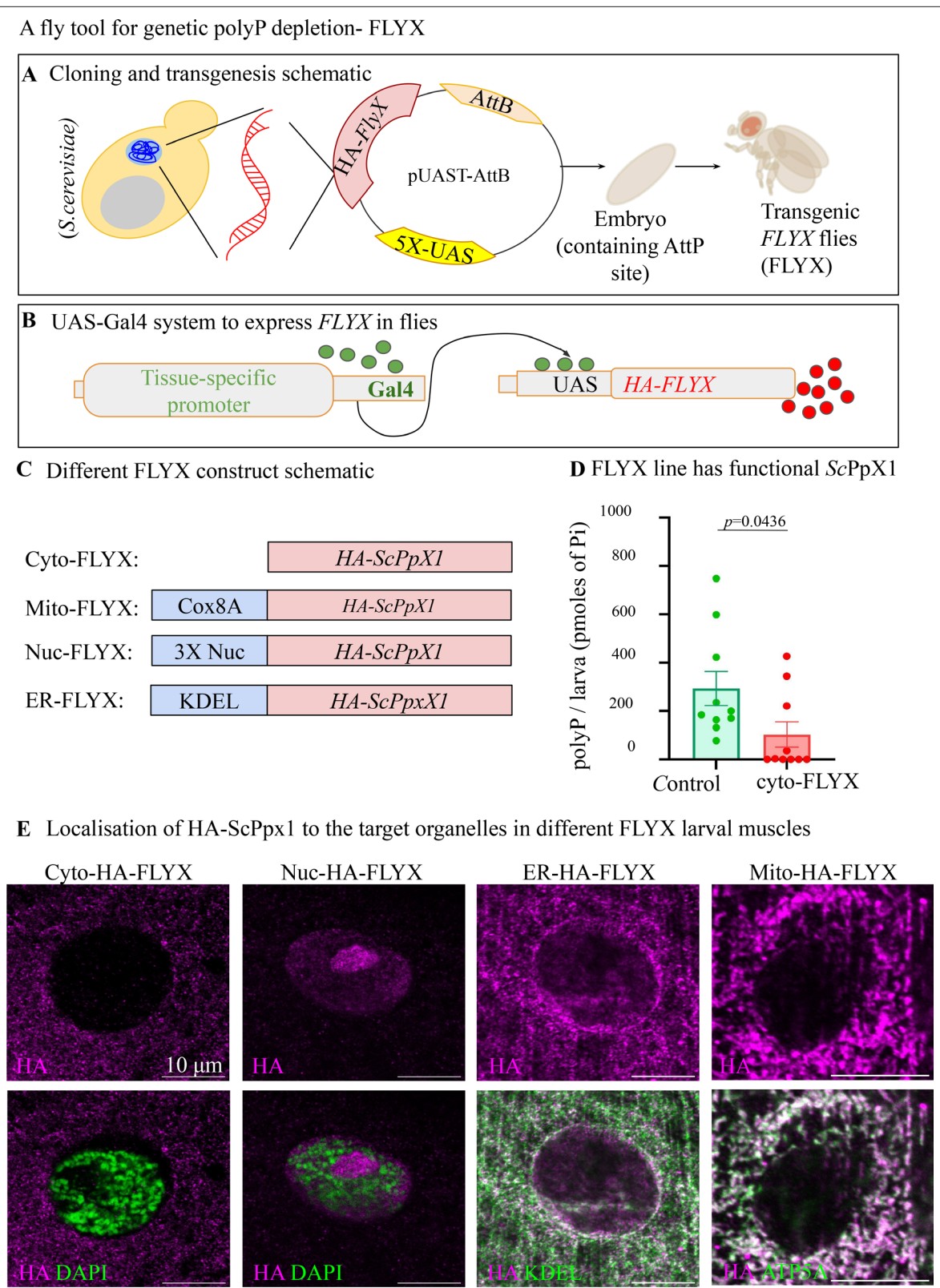

**Figure 4.** FLYX- Transgenic fly lines with Polyphosphate (polyP) depletion expressing *Sc*PpX1. (**A**) Schematic of creation of FLYX by cloning *S. cerevisiae ScPpX1* cDNA into pUAST-attB vector suitable for expression in flies, followed by injection into embryos. (**B**) Schematic of *Drosophila* UAS-GAL4-based protein expression system. (**C**) Schematic of different FLYX lines of the FLYX library- CytoFLYX, Nuc-FLYX, Mito-FLYX, and ER-FLYX. (**D**) PolyP quantification from third instar larvae of *tubulin-GAL4* driven control (AttP40) and Cyto-FLYX, N=10. Statistics: Student's *t*-test with *p*>0.001, error bar

*Figure 4 continued on next page*

*Figure 4 continued*

- s.e.m. (**E**) Localisation of HA-*Sc*PpX1 to the target organelles in different FLYX larval muscles- *DaGAL4>CytoFLYX*, stained for HA in magenta and nucleus (DAPI in green); *DaGAL4>Nuc* FLYX, stained for HA in magenta and nucleus (DAPI in green); *DaGAL4>ER* FLYX, stained for HA in magenta and ER (calnexin in green); *Mef2GAL4>Mito* FLYX, stained for HA in magenta and mitochondria (ATP5A in green). Scale bar - 10 μm.

of metamorphosis) (*Baehrecke, 1996*). In synergy, the insulin signalling pathway promotes ecdysone signalling, and shows characteristic changes in expression of genes, such as insulin receptors (*inR*), insulin-like peptides (*dilps*), *chico* (encoder of inR substrates), and *dfoxo* (insulin sensor gene) (*Shingleton et al., 2005*). Insulin signalling pathways via dFoxo also controls TOR signalling pathway effector genes, such as *4ebp* (*Koyama et al., 2013*). We tested if the accelerated eclosion caused by polyP depletion is a result of transcriptomic changes in any of these aforementioned genes. We, however, did not find a significant difference in the level of expression of genes- *ecd*, *ImpL2*, *Eip74EF*, *inR*, *chico*, *dfoxo*, *dilp2*, *tor*, *s6k*, and *4ebp* between *tubulin* >Cyto FLYX and control third instar larvae, suggesting that polyP may not directly impact Ecdysone, insulin and TOR pathways (*Figure 6—figure supplement 2A*).

To understand the cellular response to polyP depletion, we performed RNA sequencing of non-feeding time-matched third instar control and *tubulin-GAL4* >Cyto FLYX larvae. Our data covered a total of 11747 genes, and the Principal Component Analysis (PCA) segregated the two genotypes in the major axis, suggesting ubiquitous depletion of polyP changes the transcriptional landscape (*Figure 6—figure supplement 2B*). Using Gene Set Enrichment Analysis, with Gene Ontology annotations, we found a total of 262 sets of biological processes. We found translation and ribosome biogenesis pathways among the top ten upregulated biological processes in the Cyto-FLYX third instar larvae. This is interesting as polyP has been proposed to bind and modulate translation factors in other systems (*Bentley-DeSousa et al., 2018*; *Baijal et al., 2024*). A possible increase in translation could be a reason for accelerated developmental progression.

Among the gene sets downregulated in Cyto-FLYX third instar larvae, we found cell-cell adhesion and synaptic transmissions among the top ten hits. With the Cellular Component functional analyses, we also found downregulation of genes linked to neuronal projections and synapses in Cyto-FLYX larvae compared to control. Interestingly, synapses and dendritic connections are known to be pruned during larval to pupal transition as flies undergo neural circuit remodelling (*Yaniv and Schuldiner, 2016*; *Figure 6—figure supplement 3A–D*). In corroboration with the qPCR data, we did not find enrichment of genes linked to canonical ecdysone, TOR, and insulin pathways. Furthermore, we also found enrichment of negative regulators of immune response in Cyto-FLYX larvae as compared to the control. Humoral immune response in *Drosophila* is under ecdysone signalling influence, and it leads to downregulation of classical Toll and IMD pathways during larval to pupal transition. Thus, these Cyto-FLYX larvae might be susceptible to infections (*Verma and Tapadia, 2015*) (See *Supplementary file 2* for RNA sequencing analysis). Although the transcriptomic analysis from whole larvae undermines tissue-specific transcriptional changes, the wide range of impact of polyP depletion on whole larvae indicates a diverse role of polyP in organismal physiology.

## Discussion

Here, we reported methods for quantification, visualisation, and genetic manipulation of polyP in flies. To quantify polyP levels, we streamlined a colorimetric-based protocol, and for the detection of polyP in tissues, we used a GST-PPBD-based probe that specifically binds to polyP. We investigated polyP levels in various developmental stages and tissue types and found that polyP levels are spatially, temporally, and developmentally regulated. Furthermore, we created the FLYX system that allowed us to deplete polyP in a sub-cellular and tissue-specific manner. Using FLYX we observed changes in the eclosion time of flies, suggesting the importance of the temporal polyP level regulation. We also reported that the polyP in plasmatocytes is crucial for hemolymph clotting, a process analogous to human blood clotting. With the evidence of the probable evolutionarily conserved function of polyP in flies and humans, the FLYX toolkit in flies provides a handle to explore novel functions of polyP. A possible limitation of this system is the lack of an inactive *Sc*PpX1 that can deplete polyP and thus can be used as a negative control to rule out any possible effects of overexpressing the exogenous protein.

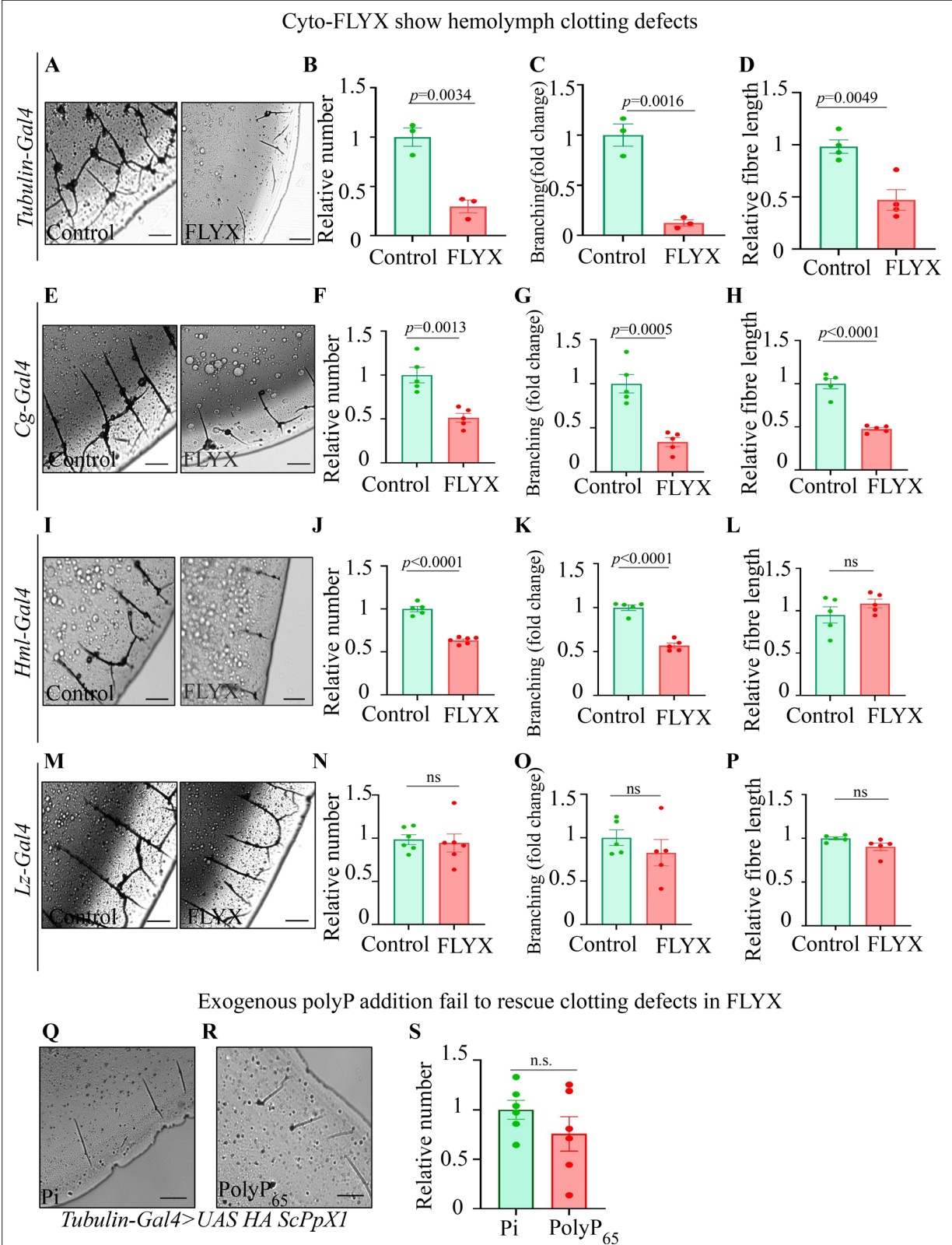

Figure 5. Genetic depletion of Polyphosphate (polyP) shows clotting defects in flies. (A–D) Clot phenotype analysis of *tubulin-GAL4* driven Cyto-FLYX (*tubulin*-FLYX). The control is *tubulin-GAL4*>AttP40. (A) Clot structure of control (AttP40) and FLYX. The scale bar is 500 pixels, and the image dimensions in pixels are 2688×2200. (B–D) Quantification of relative clot fibre number density N=3 (B), clot fibre branch point number density N=3 (C) and clot fibre length, N=4 (D) of FLYX line with respect to control. Statistics: Student's *t*-test with *p*>0.001, error bar - s.e.m. (E–P) Analysis of clot fibre number,

*Figure 5 continued on next page*

*Figure 5 continued*

branching, and length phenotype upon FLYX driven by *cg-GAL4* (**E–H**), *hml*-GAL4 (**I–L**), and *lz*-GAL4 (**M–P**) with respect to control (GAL4>AttP40). Scale bar- 500 px, Image dimensions in pixels: 2688×2200. For quantifications - N=5. Statistics: Student's *t*-test with *p*>0.001, error bar - s.e.m. (**Q–S**) Clot phenotype of *tubulin-GAL4* driven Cyto-FLYX with exogenous Pi (**Q**) and polyP$_{65}$ addition (**R**) and quantification of fibre number density (**S**). Scale bar- 500 px, Image dimensions in pixels: 2688×2200.

The online version of this article includes the following figure supplement(s) for figure 5:

**Figure supplement 1.** Polyphosphate affects hemolymph clotting.

**Figure supplement 2.** Hemolymph clotting assay.

Due to the inherent differences in polyP concentrations and chain lengths in different organisms, modifications in the polyP extraction and detection methods were required. To quantify polyP in flies, we adopted the citrate-saturated phenol-chloroform-based method, which allows the purification of all chain lengths; however, with the caveat of copurification of RNA (*Bru et al., 2016*; *Christ et al., 2020*). Since RNA can interfere with polyP quantification, we incorporated an additional RNase treatment step into our extraction protocol. PolyP is often detected using DAPI, which, when bound to polyP and excited at 405 nm, emits fluorescence with a peak at 550 nm (*Omelon et al., 2016*; *Kulakova et al., 2011*; *Aschar-Sobbi et al., 2008*; *Tanious et al., 1992*). This spectral property of DAPI-polyP has been exploited for polyP detection despite the interference from DAPI-RNA, which has a fluorescence emission maximum at 525 nm when excited at 405 nm. This interference may result in a false assessment of polyP levels, especially when polyP levels are very low, and RNAse treatment may not be efficient in digesting the complete RNA (*Saito et al., 2005*). Thus, for the biochemical detection, we used an enzyme-based method that involves enzymatic degradation of polyP, resulting in the release of Pi, which is then measured using Malachite Green (*Petitou et al., 1978*; *Hohenwallner and Wimmer, 1973*). Furthermore, to locate polyP in fly tissues, we utilised epitope-tagged PPBD, which specifically binds to polyP with a very high affinity (*Saito et al., 2005*; *Moreno-Sanchez et al., 2012*). Using these tools, we assayed the polyP levels and their localisation profiles across different fly developmental stages and in various tissue types, revealing spatiotemporal regulation of polyP in flies and

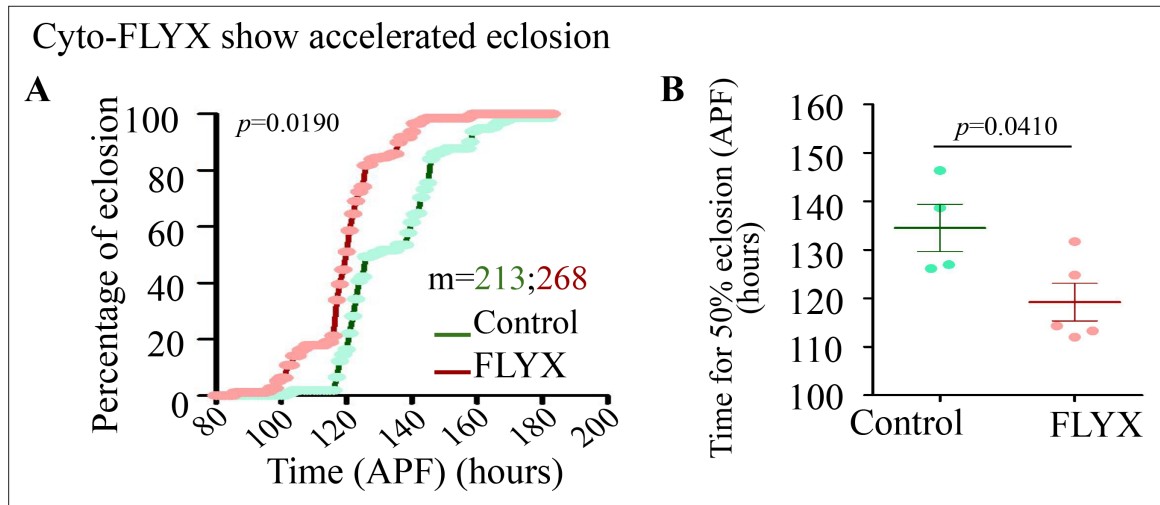

**Figure 6.** Genetic depletion of Polyphosphate (polyP) shows accelerated eclosion. (**A**) Cumulative percentage of eclosion of control and FLYX checked after white prepupa formation (APF), m=213 (control), 268 (FLYX), N=4 (control), N=5(FLYX). (**B**) Time of 50% eclosion of the control and FLYX flies- ~134 hr APF for control and 120 hr APF for FLYX, m=213 (control), 268 (FLYX), N=4 (control), N=5(FLYX). Statistics: Student's *t*-test with *p*>0.001, error bar - s.e.m. RNA sequencing of *tubulin-GAL4* driven Cyto-FLYX (tubulin>FLYX) and *tubulin-GAL4*>AttP40 control third instar non-feeding wandering age-atched larvae.

The online version of this article includes the following figure supplement(s) for figure 6:

**Figure supplement 1.** Phenotypic analysis of Cyto-FLYX w.r.t control.

**Figure supplement 2.** Transcriptomic analysis of *TubGAL4* driven Cyto-FLYX w.r.t. control.

**Figure supplement 3.** Transcriptomic analysis of *TubGAL4* driven Cyto-FLYX w.r.t. Control, GSEA, and heat map.

providing a comprehensive estimation and distribution of polyP in a metazoan organism (*Figures 1–3*, *Figure 1—figure supplement 1*, *Figure 2—figure supplement 1*, *Figure 3—figure supplement 1*).

Comparative polyP levels across different developmental stages and tissue types in metazoans have yet to be assessed, and it is tempting to predict that differential regulation of polyP levels might be conserved during the development of all multicellular organisms due to complex and differential metabolic regulation. Intriguingly, though polyP level does not change significantly during the *Drosophila* embryonic development, we observed a significant increase in polyP level in the late larval stage, just before the pupariation, followed by a gradual decrease during metamorphosis (*Figure 1C–D*). We suspect that phosphates are synthesised and stored as a reservoir in the form of polyP during the late larval stage, which is then used during metamorphosis. An assessment of polyP levels during the development of *Dictyostelium discoideum*, a model that has been used to study the origin of multicellularity, revealed a 100-fold increase in polyP levels during the transition from a single cellular vegetative state to a multicellular fruiting body. This increase in polyP is necessary for metabolic regulation during the development of *Dictyostelium*, linking the necessity of polyP to the origin of multicellularity (*Livermore et al., 2016*). Surprisingly, polyP-depleted flies (Cyto-FLYX) eclose faster (*Figure 6A–B*), uncovering a yet unknown function of polyP during metamorphosis. Intriguingly, this faster eclosion did not affect the size and weight of larvae and adults (*Figure 6—figure supplement 1A–D*). Although it is unclear why polyP depletion leads to shorter metamorphosis time, these observations could prompt several lines of phenotypic investigations to explore polyP functions.

Through our investigations, using PPBD probe, we found differential localisation of polyP in various subcellular compartments in a tissue-specific manner. In most tissues, we found polyP localisation in the form of cytosolic puncta. While we do not know the identity of these structures, similar localisation has been observed in platelet (dense granules) and mast cells (serotonin-containing granules) (*Moreno-Sanchez et al., 2012*; *Ruiz et al., 2004*) Strikingly, we observed intense PPBD staining in the nucleolus of salivary glands, muscle, crop, and wing disc cells. We also found nuclear-targeted HA-*Sc*PpX1(Nuc-FLYX) localises in the nucleoli of the nucleus of muscle cells. These data indicate a nucleolar function of polyP in some cell types. The nucleolar polyP has also been previously reported in myeloma cells and cisplatin-treated HeLa cells (*Jimenez-Nuñez et al., 2012*; *Xie et al., 2019*). Nucleolar polyP has also been observed upon forced delivery of polyP into mammalian cells or by overexpression of polyP kinase in plant and mammalian cells (*Fernandes-Cunha et al., 2018*; *Borghi et al., 2024*; *Lorenzo-Orts et al., 2019*). Since polyP can induce LLPS by its interactions with the nucleic acid and positively charged proteins, polyP may be involved in the organisation of the nucleolus, which is an LLPS organelle (*Borghi et al., 2024*; *Chawla et al., 2022*).

Overall, the polyP localisation profile using GST-PPBD staining, which uncovered spatial and temporal regulation of polyP across various *Drosophila* tissues, would accelerate the discovery of various tissue-specific functions of polyP, particularly with the use of sophisticated fly genetic tools in vivo (*Figures 2 and 3*, *Figure 3—figure supplement 1*). That the fly gut microbiota can also potentially contribute to the polyP content in fly tissues is one important consideration. In this article we allude to several observations that argue polyP is synthesized by fly tissues: (i) polyP levels remain very low during feeding stages but build up in wandering third instar larvae after feeding ceases (*Figure 1D*); (ii) PPBD staining is absent from the gut except the crop (*Figure 3—figure supplement 1O–P*); and (iii) Recent work on *C. elegans* have reported polyP presence in the worm intestines, whereas in flies, polyP is present only in the crop with the rest of the gut containing undetectable amounts. Moreover, in *C. elegans*, intestinal polyP was unaffected when worms were fed polyP-deficient bacteria (*Quarles et al., 2024*).

Recently, decreased polyP levels were found in the plasma of patients suffering from Type 1 Von Willebrand Disease (VWD) (*Montilla et al., 2012*). Moreover, mice lacking the inositol pyrophosphate synthesising enzyme IP6K1 are reported to have reduced polyP levels and show blood clotting defects (*Ghosh et al., 2013*). Despite blood clotting being the best-known function of polyP in metazoans, studies on polyP-mediated blood clotting in mammals have been done by the ex vivo addition of polyP due to the limitation of direct genetic manipulations of polyP synthesis in platelets. Using FLYX- a transgenic fly system of polyP depletion, we show that polyP in plasmatocytes is crucial for hemolymph clotting (*Figure 5A-S*). Studies in humans revealed that polyP localises inside granules in platelets, which, over the past two decades, have been shown to aid in mammalian blood clotting (*Ruiz et al., 2004*). Given our observation of the peripheral arrangement of polyP granules, we

propose that polyP is either required for secretion, modification of clotting factors in plasmatocytes or contributes to the localised increase in polyP levels at the clotting site (*Figure 5Q-S*). Our data also indicates that the polyP in fat bodies also contributes to hemolymph clotting - notably, the length of clot fibres becomes significantly shorter when polyP is depleted in all hemocytes as well as fat bodies together, but not when depleted only in plasmatocytes (*Figure 5H and L*). As fat bodies are known to secrete hemolymph clotting factors, including proteins like Fondue, which gets incorporated in the clots, polyP may modify or promote the secretion or activation of Fondue or any other factor crucial for fibre lengthening (*Lindgren et al., 2008*; *Schmid et al., 2019*). Given conserved mechanisms of hemolymph and blood clotting, modelling polyP studies in flies can help understand the regulation, release and function of polyP during hemolymph and blood clotting.

The change in the whole larval transcriptome of Cyto-FLYX reiterates the myriad possibilities of polyP functions in metazoans. We performed GSEA due to the intra-sample variation (PC2=19.10%, *Figure 6—figure supplement 3B*) as well as our aim to identify major differences in cellular responses/ pathways between Cyto-FLYX and Control (*Chicco and Agapito, 2022*). The third instar larval population usually undergoes bursts of transcription related to its development and metamorphosis. Through GSEA, we found gene sets enriched for translation and ribosome biogenesis (*Figure 6— figure supplement 3A–D*). Indeed, the proteomic analysis of a Δ*ppk* mutant *E. coli*, which results in the absence of polyP, also shows a similar response (*Baijal et al., 2023*). The reason for the increase in such biological processes in the absence of polyP (either due to no polyP synthesis or polyP depletion) remains to be investigated. Earlier polyP binding assays have shown that polyP can bind to proteins associated with translation. By linking polyP depletion to both increased translational gene expression and accelerated developmental progression, we are tempted to hypothesise a mechanistic axis in which polyP restrains protein synthesis to coordinate the timing of metamorphosis. Overall, by cytoplasmic depletion of polyP with the FLYX toolkit, we demonstrated the conservation of polyP function in hemolymph/blood clotting and the importance of polyP regulation during metamorphosis.

Through transcriptomics, followed by Gene Set Enrichment Analysis, we found the impact of cytoplasmic polyP in several biological processes, including translation and ribosome biogenesis processes during the late larval stage, reiterating the myriad possibilities of polyP functions in metazoans. An unbiased phenotypic analysis following polyP depletion in various tissues and subcellular compartments using FLYX lines developed in this work would facilitate the discovery of many unknown functions of polyP. Furthermore, the use of *Drosophila* genetic epistatic studies, by combining spatial and temporal regulation of polyP with structure and function analysis, would be useful in assigning biological meaning to several metazoan polyphosphorylated proteins that have been identified in recent studies (*Azevedo et al., 2018*; *Azevedo et al., 2015*; *Neville et al., 2023*). Finally, given the recent implications of polyP in human diseases, the fly model would be a valuable and unique in vivo system to uncover the pathogenic mechanisms of polyP-linked diseases.

In summary, this study establishes *Drosophila* as a much-needed metazoan model for polyphosphate biology, revealing conserved roles in clotting and development. The tools and findings presented here will pave the ways for new investigations to uncover functions of this ancient polymer in animal biology and its implications in human diseases.

## Materials and methods
### Polyphosphate extraction

Fly samples were lysed in LETS buffer (10 mM Tris-HCl, pH 8.0, 100 mM lithium chloride, 10 mM EDTA, 0.2% SDS); 200 µl LETS buffer was added to 20 third instar larvae, and the larvae were crushed using a pestle at room temperature. To precipitate the genomic DNA, acid phenol (pH 4.5), equivalent to the volume of the lysis buffer, was added, and the samples were centrifuged at 18,000 g for 10 min at room temperature. The aqueous phase, which is devoid of genomic DNA, was transferred to a fresh vial and mixed with a double volume of chloroform. The mixture was vortexed for 3 min and centrifuged at room temperature at 18,000 g for 5 min. The aqueous phase, devoid of proteins, was collected and treated with RNase A at 37°C for 4 hr, followed by the addition of an equal volume of 1:1 phenol: chloroform mixture. The solution was vortexed for 3 min and centrifuged at room temperature at 18,000 g for 5 min. The aqueous phase was collected, and chloroform was added to it in a volume double that of the samples. The mixture was vortexed for 3 min and centrifuged at

room temperature at 18,000 g for 5 min. The aqueous phase was collected and mixed with absolute ethanol in the ratio 1:2.5 (vol/vol), followed by overnight incubation at –80°C. The next day, the polyP was precipitated by centrifugation at 18,000 g at 4°C for 30 min. Ethanol was decanted, and the transparent polyP pellet was air-dried before dissolving in water.

## Polyphosphate quantification using malachite green assay

Malachite green (MG) detects monophosphates released from polyP pools due to the addition of purified yeast exopolyphosphatase (ScPpX1). Samples with extracted polyP were divided into two parts; one part was treated with 5 µg/ml ScPpX1 for 18 hr at 37°C, while the other half acted as the untreated control. MG reagent was prepared by mixing MG (0.045% in water) and Ammonium molybdate (4.2% in 4 N HCl) in 3:1 (vol/vol) ratio and filtered through a Whatman grade 1 paper. $K_2HPO_4$ (0.1–20 nmoles) was used as a phosphate standard. Samples or standards (50 µl) were loaded into 96 healthy plates, and 200 µl of MG reagent was added per sample, followed by incubation in the dark for 15 min at room temperature. Absorbance was measured at 650 nm using the Omega PolarStar Plate Reader. PolyP content of the samples (in Pi terms) was determined by interpolation from a linear regression analysis of the $K_2HPO_4$ standard.

## Cloning of constructs for protein purification

The plasmid pTrc-HisB-*ScPpX1* was a kind gift from Adolfo Saiardi (**Lonetti et al., 2011**). N-terminal XPRS tagged PPBD (PPXc) was cloned out from pETM41-PPXc (addgene #38329, **Werner et al., 2007**) was cloned into the parent vector pGEX-6P2-GST using restriction sites BamHI and XhoI to create pGEX-6P2-GST-XPRS-PPBD (pGEX-6P2-GST-PPBD).

## Protein purification

### *Sc*Ppx1 purification

*E. coli* BL21(DE3) competent cells transformed with pTrc-HisB-*ScPpx1* were incubated as a primary culture in Luria Bertani (LB) broth containing 100 ug/ml ampicillin overnight at 37°C. The culture was diluted into 1 L LB broth and incubated at 37°C till $OD_{600}$ reached 0.8. Protein expression was induced with 1 mM IPTG for 4–6 hr at 30°C with shaking at 180 rpm. The cells were pelleted by centrifugation at 3000 g for 15 min at 4°C. The cell pellet was resuspended in 12 ml ice-cold lysis buffer (25 mM Tris-HCl pH 8.0, 100 mM NaCl). This was followed by lysis of the cells by sonication (15 min, output voltage 20, output control 30, duty cycle 30/40) using a Branson Sonifier on ice until the lysate became clear. The lysate was centrifuged at 15,000 g for 20 min at 4°C. The supernatant was transferred to Ni-NTA column (1 mL bed volume, pre-equilibrated with lysis buffer) and incubated in an end-over mixer at 4°C for 2 hr. The flow through was collected from the column under gravity at room temperature. The column was sequentially washed with ten column volumes each of wash buffer I (25 mM Tris-HCl pH 8.0, 100 mM NaCl, and 50 mM imidazole), and wash buffer II (25 mM Tris-HCl pH 7.4, 100 mM NaCl, and 75 mM imidazole), and eluted in 5 mL elution buffers each containing 25 mM Tris-HCl pH 7.4, 100 mM NaCl, and imidazole (100 mM, 150 mM, 200 mM, and 300 mM, respectively). The eluate pooled together was subjected to size exclusion chromatography on a Sephadex-200 sizing column in buffer containing 25 mM Tris-HCl pH 7.4, 100 mM NaCl to obtain isolated fractions of *Sc*Ppx1 protein and remove contaminating proteins. The purified *Sc*Ppx1 protein was divided into single-use aliquots and stored at –80°C.

### GST, GST-PPBD, and GST-PPBD^{Mut}, protein purification

*E. coli* BL21(DE3) competent cells transformed with pGEX-6P2-GST, pGEX-6P2-GST-PPBD, and pGEX-6P2-GST-PPBD^{Mut}, respectively, were incubated as the primary culture in Luria Bertani (LB) broth containing 100 ug/ml ampicillin overnight at 37°C. The culture was diluted into 500 mL LB broth and incubated at 37°C till $OD_{600}$ reached 0.8. Protein expression was induced with 1 mM IPTG for 4 hr at 37°C and shaking at 180 rpm. The cells were pelleted by centrifugation at 3000 g for 5 min at 4°C. The pellet was resuspended in 50 ml buffer A (20 mM HEPES-KOH pH 6.8, 100 mM NaCl, 2 mM EDTA, 5 mM DTT (freshly added)) and lysed by sonication (15 min, output voltage 20, output control 30, duty cycle 30/40) using a Branson Sonifier. The lysate was centrifuged at 14,000 g for 10 min at 4°C, and the supernatant was mixed with TritonX-100 to a final concentration of 0.1%. The lysate was incubated with 250 µl of a 1:1 slurry of glutathione beads (pre-equilibrated with buffer B (buffer A plus

1% Triton X-100)) at 4°C for 2.5 hr, and the beads were pelleted at 400 g for 3 min at 4°C. The beads were washed thrice with buffer C (20 mM HEPES-KOH pH 6.8, 500 mM NaCl, 2 mM EDTA, 1% Triton X-100) and again thrice with buffer B followed by one wash with PBS, with centrifugation at 400 g for 3 min at 4°C at each wash step. The protein was eluted from the beads with 200 μl elution buffer (50 mM Tris-Cl pH 8.0, 50 mM reduced glutathione, 12 μl of 1 M NaOH) and incubated overnight on a rotor at 4°C. Eluted protein in the supernatant was recovered by centrifugation at 400 g at 4°C and stored in aliquots at –80°C.

## GST-PPBD staining

Tissues were dissected in 1X TBS (Tris Buffered Saline - 50 mM Tris-HCl pH 7.6, 150 mM NaCl), and fixed in 4% paraformaldehyde for 20 min. They were then subjected to three 1X TBST (TBS with 0.2% Triton X-100) washes for 10 min each. 5% NGS (nascent goat serum) was added to the tissues as a blocking agent, and incubated for 1 hr at room temperature. The tissues were next incubated with 25 ng/μl GST or GST-PPBD (purification process mentioned in the GST and GST-PPBD Protein Purification section of methods) for 1 hr at room temperature followed by three 1X TBST washes for 10 min each. The tissues were then incubated in mouse anti-GST antibodies overnight at 4°C. The next day, the tissues were washed thrice with 1X TBST and incubated with goat anti-mouse antibody conjugated with Alexa555 dye at room temperature for 2 hr. Following incubation, the tissues were washed thrice with 1X TBST and incubated with 100 μM DAPI to stain nuclei for 15 min at room temperature. DAPI staining was followed by a final step of three washes with 1X TBST before mounting the dissected tissues on slides for imaging. Images were acquired with a Leica STELLARIS 5 confocal microscope.

For colocalisation with the nucleolus, Rabbit anti-Fibrillarin antibody was incubated with the primary Mouse anti-GST antibodies overnight at 4°C. The next day, the tissues were washed thrice with 1X TBST and incubated with goat anti-mouse antibody conjugated with Alexa555, and goat anti-rabbit antibody conjugated with Alexa488 dye at room temperature for 2 hr. This was followed by a final step of three washes with 1X TBST before mounting the dissected tissues on slides for imaging. Images were acquired with a Leica STELLARIS 5 confocal microscope. Figures of *Figure 3—figure supplement 1A–B* were imaged with a Nikon microscope CSU-W1.

## Creation of FLYX - transgenic *Sc*PpX1 flies

The *ScPpX1* gene sequence from pTrc-HisB-*ScPpX1* was codon optimised for *Drosophila melanogaster* in the open access platform Benchling. The codon-optimised *ScPpX1* sequence (See *Supplementary file 1*) was tagged with HA epitope (TATCCGTATGATGTTCCGGATTATGCA) in the N-terminal region, and flanked by restriction sites EcoRI and XbaI. The entire fragment was synthesised by GeneScript. The HA-*ScPpX1* was cloned out from the carrier vector using restriction sites EcoR1 and Xba1 into the fly expression vector pUAST-attB. The plasmid was then injected into embryos for site-directed transgenesis in the second chromosome using the *y; AttP40* flies. Two independent lines (Cyto-FLYX) were generated, and one was used in the experiments of this study.

For targeted *Sc*PpX1 expression, the codon-optimised *ScPpX1* sequence (See *Supplementary file 1*) was tagged with HA epitope (TATCCGTATGATGTTCCGGATTATGCA) in the N-terminal region following the 3XNLS (nuclear localisation signal) for Nuc-FLYX, Cox8A signal sequence for Mito-FLYX, and the KDEL sequence for ER-FLYX. The gene inserts (See *Supplementary file 1*) were all flanked by restriction sites EcoRI and XbaI. The entire fragment was synthesised by GeneScript. The HA-*ScPpX1* was cloned out from the carrier vector using restriction sites EcoR1 and Xba1 into the fly expression vector pUAST-attB. The plasmid was then injected into embryos for site-directed transgenesis in the second chromosome using the *y; AttP40* flies. Two independent lines were generated, and one was used in the experiments of this study.

For expression in the gonads, the Cyto-FLYX construct was subcloned into the pUASp vector, and three independent lines based on p-element insertions in the second chromosome were identified and retrieved.

## Immunohistochemistry

Larval filae were prepared for staining muscle cells for HA expression in GAL4-driven FLYX lines in 1X TBS (Tris Buffered Saline - 50 mM Tris-HCl pH 7.6, 150 mM NaCl), and fixed in 4% paraformaldehyde for 20 min. They were then subjected to three 1X TBST (TBS with 0.2% Triton X-100) washes for 10

min each. 5% NGS (nascent goat serum) was added to the tissues as a blocking agent, and incubated for 1 hr at room temperature. For Cyto-FLYX and Nuc-FLYX, to stain HA, we used mouse anti-HA primary antibody whereas, for ER-FLYX and Mito-FLYX, to stain HA, we used rabbit anti-HA primary antibody, mouse anti-KDEL primary antibody (ER-F:YX) and mouse anti-ATP5A (Mito-FLYX) primary antibody. The tissues were incubated overnight at 4°C. The next day, the tissues were washed thrice with 1X TBST and incubated with goat anti-mouse antibody conjugated with Alexa555, and goat anti-rabbit antibody conjugated with Alexa488 dye at room temperature for 2 hr. The tissues were washed thrice with 1X TBST buffer. For Cyto-FLYX and Nuc-FLYX, after the secondary antibody incubation and washing, tissues were incubated with 100 µM DAPI to stain nuclei for 15 min at room temperature. This was followed by a final step of three washes with 1X TBST before mounting the dissected tissues on slides for imaging. Images were acquired with a Leica STELLARIS 5 confocal microscope.

### Hemocyte adherence and immunostaining

For adherence, the hemolymph was collected from 10 larvae in 15 µl TBS and allowed to adhere on a glass slide at room temperature for 30 min - the method adopted by *Handke et al., 2013*. The supernatant was taken out carefully, and the adhered cells were fixed using 4% PFA for 20 min. This was followed by immunostaining with GST-PPBD using the GST-PPBD staining procedure mentioned above.

For treatment of hemocytes with *Sc*PpX1 and heat inactivated *Sc*PpX1, hemocytes were subjected to three 1X TBST (TBS with 0.2% Triton X-100) washes for 10 min each, and then incubated with 5 ug/ml of *Sc*PpX1 in 1X TBS or heat inactivated *Sc*PpX1 in 1X TBS (incubated protein at 65°C for 30 min) at 37°C for 2 hr. The control untreated hemocytes were also given the same incubation at 37°C for 2 hr. This was followed by three washes and immunostaining with GST-PPBD using the GST-PPBD staining procedure mentioned above.

### Assessment of clot fibre characteristics

For assessment of the clot fibre parameters - number, branching, and length- 20 larvae were collected, washed in 1X TBS, and dried on tissue paper. The dorsal side mouth hook region was gently torn to let only the hemolymph ooze out. For *Figure 5—figure supplement 1* and *Figure 5—figure supplement 2*, the hemolymph was collected and divided into two sets of 2 µl drops. 2 µl hemolymph was mixed with either 1 µl water or 1 µl Pi (0.8 nmole) as control, and 1 µl PolyP (PolyP$_{14}$, PolyP$_{65}$, PolyP$_{130}$, each containing 0.8 nmole in Pi terms) as test samples. For *Figure 5—figure supplement 2*, and *Figure 5*, the hemolymph was collected, and 2 µl was mixed with 1 µl of water in each of the cases tested. The slide was incubated at room temperature (24°C) for 25–30 min to allow evaporation. Post-incubation, clot fibres were imaged using an Olympus BX53 upright microscope with a 10X objective lens fitted with a Retiga-R6 camera. Post imaging, the number of fibres, fibre branch points, fibre length, and circumference of the clot were measured manually using ImageJ analysis software. The number of fibres, branch points, and length of fibres were then normalised to the circumference of the respective clot. The mean of the normalised data sets of the control was analysed. Every data set is represented as relative to the mean of the normalised data of the control.

### Eclosion kinetic study

To study pupal eclosion kinetics, flies were allowed to lay eggs for 12 hr, after which 150 first instar larvae were collected in a span of 30 min to have a synchronous culture and grown at 25°C. White prepupa were collected after 6 days and staged again. The time of collection of white prepupa was set to T=0. The time of eclosion of flies was noted. A total of 213 control flies, and 268 FLYX were scored for eclosion in three independent sets. Only plates from where at least 25 flies eclosed in a day were considered for the study.

## Synthesis of PolyP$_{100}$-2X-FITC

### Polydisperse diamine-linked polydisperse PolyP$_{100}$

**Chemical structure 1.** Polydisperse diamine-linked polydisperse PolyP$_{100}$.

The synthesis was carried out following a synthetic procedure described by Fernandes-Cunha et al. (**Kornberg et al., 1999**).

Polydisperse polyP$_{100}$ (50 mg, ~5.9 µg, 1.0 eq.) was dissolved in MES buffer (1 M, 0.5 mL). 2,2'-(Ethylendioxy)diethylamin (20 eq.) and EDAC (20 eq.) were added. The pH of the suspension was adjusted to 5.7 with HCl (2 M), resulting in a clear solution. The reaction mixture was heated to 60°C for 2 hr. Subsequently, the pH was increased to pH = 10 with NaOH (2.5 M) and the crude product was precipitated in cold ethanol (35 mL), washed with cold ethanol (3x), and dried. Polydisperse diamine-linked polyP$_{100}$ (48 mg) was obtained as white solid.

$^{1}$**H-NMR** (400 MHz, D$_2$O) $\delta$=3.90–3.65 (m, 16H), 3.31–3.24 (m, 4H), 3.21–3.12 (m, 4H) ppm. $^{31}$**P{$^{1}$H}-NMR** (162 MHz, D$_2$O) δ = –0.45 - –1.08 (m, 2P), –21.60 - –22.30 (m, ~120) ppm.

### Bis-FITC-labeled PolyP$_{100}$ (PolyP$_{100}$-2X-FITC)

**Chemical structure 2.** Bis-FITC-labeled PolyP$_{100}$ (PolyP$_{100}$-2X-FITC).

The synthesis was performed according to a procedure from Fernandes-Cunha et al. (**Kornberg et al., 1999**).

Polydisperse diamine-linked polyP$_{100}$ (40 mg, ~4 µmol, 1.0 eq.) was dissolved in 0.1 M carbonate buffer (pH 9.0, 0.6 mL) and fluorescein 5 (6)-isothiocyanate (4.0 eq.) was added. The reaction mixture was stirred at r.t. overnight before it was precipitated in cold ethanol (35 mL) and washed with cold ethanol (3x). The crude product was dissolved in alkaline water (pH ~8) and purified by manual size exclusion chromatography (Sephadex 25). Bis-FITC-labeled polyP$_{100}$ (15 mg) was obtained as an orange solid.

$^{1}$**H-NMR** (400 MHz, D$_2$O) $\delta$=7.80–7.77 (m, 2H), 7.64 (dd, J=8.2, 2.2 Hz, 2 H), 7.43 (d, $J$=8.2 Hz, 2H), 7.36–7.29 (m, 4H), 6.72–6.66 (m, 8H), 3.85–3.64 (m, 14H), 3.55–3.08 (m, 8H), 2.98–3.89 (m, 2 H) ppm. $^{31}$**P{$^{1}$H}-NMR** (162 MHz, D$_2$O) δ = –0.62 - –0.96 (m, 2P), –21.58 - –21.94 (m, ~120P) ppm.

## Microscale thermophoresis

2.5 nM polyP$_{100}$-2X FITC was titrated against sixteen serially diluted concentrations of each of the proteins GST, GST-PPBD, and GST-PPBD$^{Mut}$, starting from an initial protein concentration of 5 μM. Microscale Thermophoresis was performed in Instrument Nano Temper, and the normalised fluorescence units (F$_{norm}$) and the fraction of polyP$_{100}$-2X FITC bound were plotted. From the normalised fluorescence units, the binding affinity of GST, GST-PPBD, and GST-PPBD$^{Mut}$ against polyP were determined.

## Embryo collection

Flies acclimatised on the grape juice plates for 24 hr were used for egg laying for 2 hr on a fresh grape juice plate. Every 2 hr after egg-laying, we collected the embryos for polyP extraction. The eggs were incubated at 25°C for 18 hr. We used 150 embryos for each set in our experiment.

## RNA extraction and qRT-PCR

RNA isolation using TRIzol (Ambion Life Tech—15596018) method. cDNA conversion for 1 μg of RNA was carried out using a cDNA conversion kit (Thermo Fisher—4368814). qPCR was carried out in 96-well plates in three technical replicates for each of the three biological replicate sets. Each biological replicate set had five time-matched non-feeding wandering third instar larvae.

## RNA transcriptome analysis

Sets of five larvae were subjected to RNA isolation using TRIzol (Ambion life tech—15596018) method. The RNA library was prepared using NEBNext Ultra II Directional RNA Library Prep with Sample Purification Beads. Downstream analyses were done in the condo (23.7.4) and R (4.3.2) environments. Briefly, raw reads were pair-end pseudo-aligned to the annotated *Drosophila melanogaster* genome (BDGP6.32) dataset using Kallisto (0.48.0) (*Bray et al., 2016*) and imported to the R environment. Uniquely mapped reads were further filtered and normalised using the Trimmed mean of M-values (TMM) (*Robinson and Oshlack, 2010*) method of EdgeR/limma (4.0.3) Bioconductor library (*Robinson et al., 2010*).

Then, Gene Set Enrichment Analysis (GSEA) (*Subramanian et al., 2005*) was performed using the clusterProfiler (4.10.1) package (*Wu et al., 2021*). In essence, gseGO functions were used to identify the GSEA of Gene Ontology for Biological Processes (BP), Molecular Functions (MF), and Cellular Components (CC).

## Code availability

The code is available in *Supplementary file 1*.

## Graph and statistical analysis

All the graphs and statistical analysis were done using GraphPad Prism software. The statistical tests used are mentioned in the respective figure legends.

## Acknowledgements

We thank Adolfo Saiardi for his kind gift of plasmid pTrc-HisB-ScPpX1. We also thank T Shiba for generously sharing polyP of different chain lengths -polyP (PolyP$_{14}$, PolyP$_{65}$, PolyP$_{130}$). We thank Shubham Kumar Agrawal for help with the purification of ScPpX1 and Manisha Mallick for assistance PPX and PPBD binding experiments. We acknowledge Debaditya De, Aditya Rane, and Manasa Chanduri for the construction of the ScPpX1 and GST-PPBD expression plasmids. We thank Rakesh Mahato for assisting in the MST experiments and Dr. Pankaj Suman for sharing the MST set up at the National Institute of Animal Biotechnology, India. We thank Lolitika Mandal for sharing HmlGAL4>UAS GFP fly lines with us. We thank Aprotim Mazumder for sharing anti-Fibrillarin antibodies. We thank Vipin Agarwal for sharing the SEC setup for protein purification. We thank Dr. Deepti Trivedi and NCBS fly facility for injections and Dr. Awadhesh Pandit and his team at NCBS for RNA library preparation and RNA sequencing. We thank Kalyaneswar Mandal, Sonal Nagarkar Jaiswal, and members of the Laboratory of Cell Signalling CDFD and MJ lab TIFRH for discussions and valuable feedback. We

thank Anand T Vaidya, Vinay Bulusu, and Oguz Kanca for the critical reading, comments and valuable suggestions on the manuscript.

## Additional information

### Funding

| Funder | Grant reference number | Author |
|---|---|---|
| Department of Biotechnology, Ministry of Science and Technology, India | IC12025(11)/2/2020-ICD-DBT | Rashna Bhandari Manish Jaiswal |
| Deutsche Forschungsgemeinschaft | 445698446 | Henning J Jessen |
| Council of Scientific and Industrial Research, India | Shyama Prasad Mukherjee Fellowship | Jayashree S Ladke |
| Infosys Foundation | Infosys fellow (Leading Edge TG/(R-11)/09/ | Sunayana Sarkar |
| Department of Atomic Energy, Government of India | Project Identification No. RTI4007 | Manish Jaiswal |
| Science and Engineering Research Board | CRG/2020/003275 | Manish Jaiswal |
| Department of Biotechnology, Ministry of Science and Technology, India | BT/PR32873/BRB/10/1850/2020 | Manish Jaiswal |
| Department of Biotechnology, Ministry of Science and Technology, India | BT/RLF/Re-entry/06/2016 | Manish Jaiswal |
| Deutsche Forschungsgemeinschaft | CIBSS-EXC-2189-Project ID 390939984 | Henning J Jessen |

The funders had no role in study design, data collection and interpretation, or the decision to submit the work for publication.

### Author contributions

Sunayana Sarkar, Conceptualization, Data curation, Formal analysis, Validation, Investigation, Visualization, Methodology, Writing – original draft, Writing – review and editing; Harsha Sharma, Validation, Investigation, Methodology; SK Yasir Hosen, Data curation, Formal analysis, Investigation; Jayashree S Ladke, Methodology, Writing – review and editing; Sandra Moser, Resources, Writing – review and editing; Deepa Balasubramanian, Sreejith Raran-Kurussi, Methodology; Henning J Jessen, Resources, Funding acquisition, Project administration, Writing – review and editing; Rashna Bhandari, Conceptualization, Supervision, Funding acquisition, Project administration, Writing – review and editing; Manish Jaiswal, Conceptualization, Resources, Supervision, Funding acquisition, Writing – original draft, Project administration, Writing – review and editing

### Author ORCIDs

Sunayana Sarkar ⓘ https://orcid.org/0009-0004-9001-2247
Rashna Bhandari ⓘ https://orcid.org/0000-0003-3101-0204
Manish Jaiswal ⓘ https://orcid.org/0000-0001-8360-8289

Reviewer #2 (Public review): https://doi.org/10.7554/eLife.104841.3.sa1
Reviewer #3 (Public review): https://doi.org/10.7554/eLife.104841.3.sa2
Author response https://doi.org/10.7554/eLife.104841.3.sa3

# Additional files

## Supplementary files
MDAR checklist

Supplementary file 1. Supporting information.

Supplementary file 2. RNA sequencing data.

## Data availability
Transcriptomics data is provided in *Supplementary file 2*.

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
