## [Editor Report · eLife Assessment]

Studying the biological roles of polyphosphates in metazoans has been a longstanding challenge to the field given that the polyP synthase has yet to be discovered in metazoans. This **important** study capitalizes on the sophisticated genetics available in the Drosophila system and uses a combination of methodologies to start to tease apart how polyphosphate participates in Drosophila development and in the clotting of Drosophila hemolymph. The data validating one of these tools (cyto-FLYX) are **solid** and well-documented and they will open up a field of research into the functional roles of polyP in a metazoan model. The other tools for tissue specific knockdown of polyP (Mito-FLYX, ER-FLYX, and Nuc-FLYX) have not yet been validated but will be invaluable to the field when they are.

---

## [Referee Report · Reviewer #2 (Public review)]

Summary:

The authors of this paper note that although polyphosphate (polyP) is found throughout biology, the biological roles of polyP have been under-explored, especially in multicellular organisms. The authors created transgenic Drosophila that expressed a yeast enzyme that degrades polyP, targeting the enzyme to different subcellular compartments (cytosol, mitochondria, ER, and nucleus, terming these altered flies Cyto-FLYX, Mito-FLYX, etc.). The authors show the localization of polyP in various wild-type fruit fly cell types and demonstrate that the targeting vectors did indeed result in expression of the polyP degrading enzyme in the cells of the flies. They then go on to examine the effects of polyP depletion using just one of these targeting systems (the Cyto-FLYX). The primary findings from depletion of cytosolic polyP levels in these flies is that it accelerates eclosion and also appears to participate in hemolymph clotting. Perhaps surprisingly, the flies seemed otherwise healthy and appeared to have little other noticeable defects. The authors use transcriptomics to try to identify pathways altered by the cyto-FLYX construct degrading cytosolic polyP, and it seems likely that their findings in this regard will provide avenues for future investigation. And finally, although the authors found that eclosion is accelerated in pupae of Drosophila expressing the Cyto-FLYX construct, the reason why this happens remains unexplained.

Strengths:

The authors capitalize on the work of other investigators who had previously shown that expression of recombinant yeast exopolyphosphatase could be targeted to specific subcellular compartments to locally deplete polyP, and they also use a recombinant polyP binding protein (PPBD) developed by others to localize polyP. They combine this with the considerable power of Drosophila genetics to explore the roles of polyP by depleting it in specific compartments and cell types to tease out novel biological roles for polyP in a whole organism. This is a substantial advance.

Weaknesses:

Page 4 of Results (paragraph 1): I'm a bit concerned about the specificity of PPBD as a probe for polyP. The authors show that the fusion partner (GST) isn't responsible for the signal, but I don't think they directly demonstrate that PPBD is binding only to polyP. Could it also bind to other anionic substances? A useful control might be to digest the permeabilized cells and tissues with polyphosphatase prior to PPBD staining, and show that the staining is lost.

In the hemolymph clotting experiments, the authors collected 2 ul of hemolymph and then added 1 ul of their test substance (water or a polyP solution). They state that they added either 0.8 or 1.6 nmol polyP in these experiments (the description in the Results differs from that of the Methods). I calculate this will give a polyP concentration of 0.3 or 0.6 mM. This is an extraordinarily high polyP concentration, and is much in excess of the polyP concentrations used in most of the experiments testing the effects of polyP on clotting of mammalian plasma. Why did the authors choose this high polyP concentration? Did they try lower concentrations? It seems possible that too high a polyP concentration would actually have less clotting activity than the optimal polyP concentration.

In the revised version of the manuscript, the authors have productively responded to the previous criticisms. Their new data show stronger controls regarding the specificity of PPBD with regard to its interaction with polyP. The authors also have repeated their hemolymph clotting experiments with lower polyP concentrations, which are likely to be more physiological.

---

## [Referee Report · Reviewer #3 (Public review)]

Summary:

Sarkar, Bhandari, Jaiswal and colleagues establish a suite of quantitative and genetic tools to use *Drosophila melanogaster* as a model metazoan organism to study polyphosphate (polyP) biology. By adapting biochemical approaches for use in *D. melanogaster*, they identify a window of increased polyP levels during development. Using genetic tools, they find that depleting polyP from the cytoplasm alters the timing of metamorphosis, accelerationg eclosion. By adapting subcellular imaging approaches for *D. melanogaster*, they observe polyP in the nucleolus of several cell types. They further demonstrate that polyP localizes to cytoplasmic puncta in hemocytes, and further that depleting polyP from the cytoplasm of hemocytes impairs hemolymph clotting. Together, these findings establish *D. melanogaster* as a tractable system for advancing our understanding of polyP in metazoans.

Strengths:

• The FLYX system, combining cell type and compartment-specific expression of ScPpx1, provides a powerful tool for the polyP community.

• The finding that cytoplasmic polyP levels change during development and affect the timing of metamorphosis is an exciting first step in understanding the role of polyP in metazoan development, and possible polyP-related diseases.

• Given the significant existing body of work implicating polyP in the human blood clotting cascade, this study provides compelling evidence that polyP has an ancient role in clotting in metazoans.

Limitations:

• While the authors demonstrate that HA-ScPpx1 protein localizes to the target organelles in the various FLYX constructs, the capacity of these constructs to deplete polyP from the different cellular compartments is not shown. This is an important control to both demonstrate that the GTS-PPBD labeling protocol works, and also to establish the efficacy of compartment-specific depletion. While not necessary to do for all the constructs, it would be helpful to do this for the cyto-FLYX and nuc-FLYX.

• The cell biological data in this study clearly indicates that polyP is enriched in the nucleolus in multiple cell types, consistent with recent findings from other labs, and also that polyP affects gene expression during development. Given that the authors also generate the Nuc-FLYX construct to deplete polyP from the nucleus, it is surprising that they test how depleting cytoplasmic but not nuclear polyP affects development. However, providing these tools is a service to the community, and testing the phenotypic consequences of all the FLYX constructs may arguably be beyond the scope of this first study.

Editors' note: The authors have satisfactorily responded to our most major concerns related to the specificity of PPDB and the physiological levels of polyPs in the clotting experiments. We also recognise the limitations related to the depletion of polyP in other tissues and hope that these data will be made available soon.

---

## [Author Response]

The following is the authors’ response to the original reviews.

**Public Reviews:**

**Reviewer #1 (Public review):**
Polymers of orthophosphate of varying lengths are abundant in prokaryotes and some eukaryotes, where they regulate many cellular functions. Though they exist in metazoans, few tools exist to study their function. This study documents the development of tools to extract, measure, and deplete inorganic polyphosphates in *Drosophila*. Using these tools, the authors show:(1) That polyP levels are negligible in embryos and larvae of all stages while they are feeding. They remain high in pupae but their levels drop in adults.(2) That many cells in tissues such as the salivary glands, oocytes, haemocytes, imaginal discs, optic lobe, muscle, and crop, have polyP that is either cytoplasmic or nuclear (within the nucleolus).(3) That polyP is necessary in plasmatocytes for blood clotting in *Drosophila*.(4) That ployP controls the timing of eclosion.The tools developed in the study are innovative, well-designed, tested, and well-documented. I enjoyed reading about them and I appreciate that the authors have gone looking for the functional role of polyP in flies, which hasn't been demonstrated before. The documentation of polyP in cells is convincing as its role in plasmatocytes in clotting.

We sincerely thank the reviewer for their encouraging assessment and for recognizing both the innovation of the FLYX toolkit and the functional insights it enables. Their remarks underscore the importance of establishing *Drosophila* as a tractable model for polyP biology, and we are grateful for their constructive feedback, which further strengthened the manuscript.

Its control of eclosion timing, however, could result from non-specific effects of expressing an exogenous protein in all cells of an animal.

We now explicitly state this limitation in the revised manuscript (p.16, l.347–349). The issue is that no catalytic-dead *Sc*PpX1 is available as a control in the field. We plan to generate such mutants through systematic structural and functional studies and will update the FLYX toolkit once they are developed and validated. Importantly, the accelerated eclosion phenotype is reproducible and correlates with endogenous polyP dynamics.

The RNAseq experiments and their associated analyses on polyP-depleted animals and controls have not been discussed in sufficient detail. In its current form, the data look to be extremely variable between replicates and I'm therefore unsure of how the differentially regulated genes were identified.

We thank the reviewer for pointing out the lack of clarity. We have expanded our RNAseq analysis in the revised manuscript (p.20, l.430–434). Because of inter-sample variation (PC2 = 19.10%, Fig. S7B), we employed Gene Set Enrichment Analysis (GSEA) rather than strict DEG cutoffs. This method is widely used when the goal is to capture pathway-level changes under variability (1). We now also highlight this limitation explicitly (p.20, l.430–432) and provide an additional table with gene-specific fold change (See Supplementary Table for RNA Sequencing Sheet 1). Please note that we have moved RNAseq data to Supplementary Fig. 7 and 8 as suggested in the review.

It is interesting that no kinases and phosphatases have been identified in flies. Is it possible that flies are utilising the polyP from their gut microbiota? It would be interesting to see if these signatures go away in axenic animals.

This is an interesting possibility. Several observations argue that polyP is synthesized by fly tissues: (i) polyP levels remain very low during feeding stages but build up in wandering third instar larvae after feeding ceases; (ii) PPBD staining is absent from the gut except the crop (Fig. S3O–P); (ii) In *C. elegans*, intestinal polyP was unaffected when worms were fed polyP-deficient bacteria (2); (iv) depletion of polyP from plasmatocytes alone impairs hemolymph clotting, which would not be expected if gut-derived polyP were the major source and may have contributed to polyP in hemolymph. Nevertheless, we agree that microbiota-derived polyP may contribute, and we plan systematic testing in axenic flies in future work.

**Reviewer #2 (Public review):**
Summary:The authors of this paper note that although polyphosphate (polyP) is found throughout biology, the biological roles of polyP have been under-explored, especially in multicellular organisms. The authors created transgenic *Drosophila* that expressed a yeast enzyme that degrades polyP, targeting the enzyme to different subcellular compartments (cytosol, mitochondria, ER, and nucleus, terming these altered flies Cyto-FLYX, Mito-FLYX, etc.). The authors show the localization of polyP in various wild-type fruit fly cell types and demonstrate that the targeting vectors did indeed result in the expression of the polyP degrading enzyme in the cells of the flies. They then go on to examine the effects of polyP depletion using just one of these targeting systems (the Cyto-FLYX). The primary findings from the depletion of cytosolic polyP levels in these flies are that it accelerates eclosion and also appears to participate in hemolymph clotting. Perhaps surprisingly, the flies seemed otherwise healthy and appeared to have little other noticeable defects. The authors use transcriptomics to try to identify pathways altered by the cyto-FLYX construct degrading cytosolic polyP, and it seems likely that their findings in this regard will provide avenues for future investigation. And finally, although the authors found that eclosion is accelerated in the pupae of *Drosophila* expressing the Cyto-FLYX construct, the reason why this happens remains unexplained.Strengths:The authors capitalize on the work of other investigators who had previously shown that expression of recombinant yeast exopolyphosphatase could be targeted to specific subcellular compartments to locally deplete polyP, and they also use a recombinant polyP-binding protein (PPBD) developed by others to localize polyP. They combine this with the considerable power of *Drosophila* genetics to explore the roles of polyP by depleting it in specific compartments and cell types to tease out novel biological roles for polyP in a whole organism. This is a substantial advance.

We are grateful to the reviewer for their thorough and thoughtful evaluation. Their balanced summary of our work, recognition of the strengths of our genetic tools, and constructive suggestions have been invaluable in clarifying our experiments and strengthening the conclusions.

Weaknesses:Page 4 of the Results (paragraph 1): I'm a bit concerned about the specificity of PPBD as a probe for polyP. The authors show that the fusion partner (GST) isn't responsible for the signal, but I don't think they directly demonstrate that PPBD is binding only to polyP. Could it also bind to other anionic substances? A useful control might be to digest the permeabilized cells and tissues with polyphosphatase prior to PPBD staining and show that the staining is lost.

To address this concern, we have done two sets of experiments:

(1) We generated a PPBD mutant (GST-PPBD^Mut^). We establish that GST-PPBD binds to polyP-2X FITC, whereas GST-PPBD^Mut^ and GST do not bind polyP_100_-2X FITC using *Microscale Thermophoresis*. We found that, unlike the punctate staining pattern of GST-PPBD (wild-type), GST-PPBD^Mut^ does not stain hemocytes. This data has been added to the revised manuscript (Fig. 2B-D, p.8, l.151–165).

(2) A study in *C. elegans* by Quarles et.al has performed a similar experiment, suggested by the reviewer. In that study, treating permeabilized tissues with polyphosphatase prior to PPBD staining resulted in a decrease of PPBD-GFP signal from the tissues (2). We also performed the same experiment where we subjected hemocytes to GST-PPBD staining with prior incubation of fixed and permeabilised hemocytes with *Sc*PpX1 and heat-inactivated *Sc*PpX1 protein. We find that both staining intensity and the number of punctae are higher in hemocytes left untreated and in those treated with heat-inactivated *Sc*PpX1. The hemocytes pre-treated with *Sc*PpX1 showed reduced staining intensity and number of punctae. This data has been added to the revised manuscript (Fig. 2E-G, p.8, l.166-172).

Further, Saito et al. reported that PPBD binds to polyP in vitro, as well as in yeast and mammalian cells, with a high affinity of ~45µM for longer polyP chains (35 mer and above) (3). They also show that the affinity of PPBD with RNA and DNA is very low. Furthermore, PPBD could detect differences in polyP labeling in yeasts grown under different physiological conditions that alter polyP levels (3). Taken together, published work and our results suggest that PPBD specifically labels polyP.

In the hemolymph clotting experiments, the authors collected 2 ul of hemolymph and then added 1 ul of their test substance (water or a polyP solution). They state that they added either 0.8 or 1.6 nmol polyP in these experiments (the description in the Results differs from that of the Methods). I calculate this will give a polyP concentration of 0.3 or 0.6 mM. This is an extraordinarily high polyP concentration and is much in excess of the polyP concentrations used in most of the experiments testing the effects of polyP on clotting of mammalian plasma. Why did the authors choose this high polyP concentration? Did they try lower concentrations? It seems possible that too high a polyP concentration would actually have less clotting activity than the optimal polyP concentration.

We repeated the assays using 125 µM polyP, consistent with concentrations employed in mammalian plasma studies (4,5). Even at this lower, physiologically relevant concentration, polyP significantly enhanced clot fibre formation (Included as Fig. S5F–I, p.12, l.241–243). This reconfirms the conclusion that polyP promotes hemolymph clotting.

**Reviewer #3 (Public review):**
Summary:Sarkar, Bhandari, Jaiswal, and colleagues establish a suite of quantitative and genetic tools to use *Drosophila melanogaster* as a model metazoan organism to study polyphosphate (polyP) biology. By adapting biochemical approaches for use in *D. melanogaster*, they identify a window of increased polyP levels during development. Using genetic tools, they find that depleting polyP from the cytoplasm alters the timing of metamorphosis, accelerating eclosion. By adapting subcellular imaging approaches for *D. melanogaster*, they observe polyP in the nucleolus of several cell types. They further demonstrate that polyP localizes to cytoplasmic puncta in hemocytes, and further that depleting polyP from the cytoplasm of hemocytes impairs hemolymph clotting. Together, these findings establish *D. melanogaster* as a tractable system for advancing our understanding of polyP in metazoans.Strengths:(1) The FLYX system, combining cell type and compartment-specific expression of ScPpx1, provides a powerful tool for the polyP community.(2) The finding that cytoplasmic polyP levels change during development and affect the timing of metamorphosis is an exciting first step in understanding the role of polyP in metazoan development, and possible polyP-related diseases.(3) Given the significant existing body of work implicating polyP in the human blood clotting cascade, this study provides compelling evidence that polyP has an ancient role in clotting in metazoans.

We sincerely thank the reviewer for their generous and insightful comments. Their recognition of both the technical strengths of the FLYX system and the broader biological implications reinforces our confidence that this work will serve as a useful foundation for the community.

Limitations:(1) While the authors demonstrate that HA-ScPpx1 protein localizes to the target organelles in the various FLYX constructs, the capacity of these constructs to deplete polyP from the different cellular compartments is not shown. This is an important control to both demonstrate that the GTS-PPBD labeling protocol works, and also to establish the efficacy of compartment-specific depletion. While not necessary to do this for all the constructs, it would be helpful to do this for the cyto-FLYX and nuc-FLYX.

We confirmed polyP depletion in Cyto-FLYX using the malachite green assay (Fig. 3D, p.10, l.212–214). The efficacy of *Sc*PpX1 has also been earlier demonstrated in mammalian mitochondria (6). Our preliminary data from Mito-*Sc*PpX1 expressed ubiquitously with Tubulin-Gal4 showed a reduction in polyP levels when estimated from whole flies (See Author response image 2 below, ongoing investigation). In an independent study focusing on mitochondrial polyP depletion, we are characterizing these lines in detail and plan to check the amount of polyP contributed to the cellular pool by mitochondria using subcellular fractionation. Direct phenotypic and polyP depletion analyses of Nuc-FLYX and ER-FLYX are also being carried out, but are in preliminary stages. That there is a difference in levels of polyP in various tissues and that we get a very little subscellular fraction for polyP analysis have been a few challenging issues. This analysis requires detailed, independent, and careful analysis, and thus, we refrain from adding this data to the current manuscript.

**Author response image 2. sa3fig2:** 

Regarding the specificity, Saito et.al. reported that PPBD binds to polyP in vitro, as well as in yeast and mammalian cells with a high affinity of ~45µM for longer polyP chains (35 mer and above) (3). They also show that the affinity of PPBD with RNA and DNA is very low. Further, PPBD could reveal differences in polyP labeling with yeasts grown in different physiological conditions that can alter polyP levels. Now in the manuscript, we included following data to show specificity of PPBD:

To address this concern we have done two sets of experiments:

We generated a PPBD mutant (GST-PPBD^Mut^). Using Microscale Thermophoresis, we establish that GST-PPBD binds to polyP_100_-2X-FITC, whereas, GST-PPBD^Mut^ and GST do not bind polyP_100_-2X-FITC at all. We found that unlike the punctate staining pattern of GST-PPBD (wild-type), GST-PPBD^Mut^ does not stain hemocytes. This data has been added to the revised manuscript (Fig. 2B-D, p.8, l.151–165).

A study in *C. elegans* by Quarles et.al has performed a similar experiment suggested by the reviewer. In that study, treating permeabilized tissues with polyphosphatase prior to PPBD staining resulted in decrease of PPBD-GFP signal from the tissues (2). We also performed the same experiment where we subjected hemocytes to GST-PPBD staining with prior incubation of fixed and permeabilised hemocytes with *Sc*PpX1 and heat inactivated *Sc*PpX1 protein. We find that both intensity of staining and number of punctae are higher in hemocytes that were left untreated and the one where heat inactivated *Sc*PpX1 was added. The hemocytes pre-treated with *Sc*PpX1 showed reduced staining intensity and number of punctae. This data has been added to the revised manuscript (Fig. 2E-G, p.8, l.166-172).

(2) The cell biological data in this study clearly indicates that polyP is enriched in the nucleolus in multiple cell types, consistent with recent findings from other labs, and also that polyP affects gene expression during development. Given that the authors also generate the Nuc-FLYX construct to deplete polyP from the nucleus, it is surprising that they test how depleting cytoplasmic but not nuclear polyP affects development. However, providing these tools is a service to the community, and testing the phenotypic consequences of all the FLYX constructs may arguably be beyond the scope of this first study.

We agree this is an important avenue. In this first study, we focused on establishing the toolkit and reporting phenotypes with Cyto-FLYX. We are systematically assaying phenotypes from all FLYX constructs, including Nuc-FLYX, in ongoing studies

**Recommendations for the authors:**

**Reviewing Editor Comment:**
The reviewers appreciated the general quality of the rigour and work presented in this manuscript. We also had a few recommendations for the authors. These are listed here and the details related to them can be found in the individual reviews below.(1) We suggest including an appropriate control to show that PPBD binds polyP specifically.

We have updated the response section as follows:

(a) Highlighted previous literature that showed the specificity of PPBD.

(b) We show that the punctate staining observed by PPBD is not demonstrated by the mutant PPBD (PPBD^Mut^) in which amino acids that are responsible for polyP binding are mutated.

(c) We show that PPBD^Mut^ does not bind to polyP using Microscale Thermophoresis.

(d) We show that treatment of fixed and permeabilised hemocytes with *Sc*PpX1 reduces the PPBD staining intensity and number of punctae, as compared to tissues left untreated or treated with heat-inactivated *Sc*PpX1.

We have included these in our updated revised manuscript (Fig. 2B-G, p.8, l.151–157)

(2) The high concentration of PolyP in the clotting assay might be impeding clotting. The authors may want to consider lowering this in their assays.

We have addressed this concern in our revised manuscript. We have performed the clotting assays with lower polyP concentrations (concentrations previously used in clotting experiments with human blood and polyP). Data is included in Fig. S5F–I, p.12, l.241–243.

(3) The RNAseq study: can the authors please describe this better and possibly mine it for the regulation of genes that affect eclosion?

In our revised manuscript, we have included a broader discussion about the RNAseq analysis done in the article in both the ‘results’ and the ‘discussion’ sections, where we have rewritten the narrative from the perspective of accelerated eclosion. (p.15 l.310-335, p. 20, l.431-446).

(4) Have the authors considered the possibility that the gut microbiota might be contributing to some of their measurements and assays? It would be good to address this upfront - either experimentally, in the discussion, or (ideally) both**.**

This is an exciting possibility. Several observations argue that fly tissues synthesize polyP: (i) polyP levels remain very low during feeding stages but build up in wandering third instar larvae after feeding ceases; (ii) PPBD staining is absent from the gut except the crop (Fig. S3O–P); (iii) in *C. elegans*, intestinal polyP was unaffected when worms were fed polyP-deficient bacteria (2); (iv) depletion of polyP from plasmatocytes alone impairs hemolymph clotting, which would not be expected if gut-derived polyP were the major source and may have contributed to polyP in hemolymph. Nevertheless, microbiota-derived polyP may contribute, and we plan systematic testing in axenic flies in future work.

**Reviewer #1 (Recommendations for the authors):**
(1) While the authors have shown that the depletion tool results in a general reduction of polyP levels in Figure 3D, it would have been nice to show this via IHC. Particularly since the depletion depends on the strength of the Gal4, it is possible that the phenotypes are being under-estimated because the depletions are weak.

We agree that different Gal4 lines have different strengths and will therefore affect polyP levels and the strength of the phenotype differently.

We performed PPBD staining on hemocytes expressing *Sc*PPX; however, we observed very intense, uniform staining throughout the cells, which was unexpected. It seems like PPBD is recognizing overexpressed *Sc*PpX1. Indeed, in an unpublished study by Manisha Mallick (Bhandari lab), it was found that His-*Sc*PpX1 specifically interacts with GST-PPBD in a protein interaction assay (See Author response image 3). Due to these issues, we refrained from IHC/PPBD-based validation.

**Author response image 3. sa3fig3:** 

(2) The subcellular tools for depletion are neat! I wonder why the authors didn't test them. For example in the salivary gland for nuclear depletion?

We have addressed this question in the reviewer responses. We are systematically assaying phenotypes from all FLYX constructs, including Mito-FLYX, and Nuc-FLYX, in ongoing independent investigations. As discussed in #1, a possible interaction of *ScPpX* and PPBD is making this test a bit more challenging, and hence, they each require a detailed investigation.

(a) Does the absence of clotting defects using Lz-gal4 suggest that PolyP is more crucial in the plasmatocytoes and for the initial clotting process? And that it is dispensible/less important in the crystal cells and for the later clotting process. Or is it that the crystal cells just don't have as much polyP? The image (2E-H) certainly looks like it.

In hemolymph, the primary clot formation is a result of the clotting factors secreted from the fat bodies and the plasmatocytes. The crystal cells are responsible for the release of factors aiding in successfully hardening the soft clot initially formed. Reports suggest that clotting and melanization of the clot are independent of each other (7). Since Crystal cells do not contribute to clot fibre formation, the absence of clotting defects using LzGAL4-CytoFLYX is not surprising. Alternatively, PolyP may be secreted from all hemocytes and contribute to clotting; however, the crystal cells make up only 5% hemocytes, and hence polyP depletion in those cells may have a negligible effect on blood clotting.

Crystal cells do show PPBD staining. Whether polyP is significantly lower in levels in the crystal cells as compared to the plasmatocytes needs more systematic investigation. Image (2E-H) is a representative image of the presence of polyP in crystal cells and can not be considered to compare polyP levels in the crystal cells vs Plasmatocytes.

(b) The RNAseq analyses and data could be better presented. If the data are indeed variable and the differentially expressed genes of low confidence, I might remove that data entirely. I don't think it'll take away from the rest of the work.

We understand this concern and, therefore, in the revised manuscript, we have included a broader discussion about the RNAseq analysis done in the article in both the ‘results’ and the ‘discussion’ sections, where we have rewritten the narrative from the perspective of accelerated eclosion. (p.15 l.310-335, p. 20, l.431-446). We have also stated the limitations of such studies.

(c) I would re-phrase the first sentence of the results section.

We have re-phrased it in the revised manuscript.

**Reviewer #2 (Recommendations for the authors):**
(1) The authors created several different versions of the FLYX system that would be targeted to different subcellular compartments. They mostly report on the effects of cytosolic targeting, but some of the constructs targeted the polyphosphatase to mitochondria or the nucleus.They report that the targeting worked, but I didn't see any results on the effects of those constructs on fly viability, development, etc.There is a growing literature of investigators targeting polyphosphatase to mitochondria and showing how depleting mitochondrial polyP alters mitochondrial function. What was the effect of the Nuc-FLYX and Mito-FLYX constructs on the flies?Also, the authors should probably cite the papers of others on the effects of depleting mitochondrial polyP in other eukaryotic cells in the context of discussing their findings in flies.

We have addressed this question in the reviewer responses. We did not see any obvious developmental or viability defects with any of the FLYX lines, and only after careful investigation did we come across the clotting defects in the CytoFLYX. We are currently systematically assaying phenotypes from all FLYX constructs, including Mito-FLYX and Nuc-FLYX, in independent ongoing investigations.

We have discussed the heterologous expression of mitochondrial polyphosphatase in mammalian cells to justify the need for developing Mito-FLYX (p. 10, l. 197-200). In the discussion section, we also discuss the presence and roles of polyP in the nucleus and how Nuc-FLYX can help study such phenomena (p. 19, l. 399-407).

(2) The authors should number the pages of their manuscript to make it easier for reviewers to refer to specific pages.

We have numbered our lines and pages in the revised manuscript.

(3) Abstract: the abbreviation, "polyP", is not defined in the abstract. The first word in the abstract is "polyphosphate", so it should be defined there.

We have corrected it in the revised version.

(4) The authors repeatedly use the phrase, "orange hot", to describe one of the colors in their micrographs, but I don't know how this differs from "orange".

‘OrangeHot’ is the name of the LUT used in the ImageJ analysis and hence referred to as the colour

(5) First page of the Introduction: the phrase, "feeding polyP to αβ expression Alzheimer's model of *Caenorhabditis elegans*" is awkward (it literally means feeding polyP to the model instead of the worms).

We have revised it. (p.3, l.55-57).

(6) Page 2 of the Introduction: The authors should cite this paper when they state that NUDT3 is a polyphosphatase: https://pubmed.ncbi.nlm.nih.gov/34788624/

We have cited the paper in the revised version of the manuscript. (p.4, l. 68-70)

(7) Page 2 of Results: The authors report the polyP content in the third instar larva (misspelled as "larval") to five significant digits ("419.30"). Their data do not support more than three significant digits, though.

We have corrected it in the revised manuscript.

(8) Page 3 of Results (paragraph 1): When discussing the polyP levels in various larval stages, the authors are extracting total polyP from the larvae. It seems that at least some of the polyP may come from gut microbes. This should probably be mentioned.

This is an interesting possibility. Several observations argue that polyP is synthesized by fly tissues: (i) polyP levels remain very low during feeding stages but build up in wandering third instar larvae after feeding ceases; (ii) PPBD staining is absent from the gut except the crop (Fig. S3O–P); (ii) In *C. elegans*, intestinal polyP was unaffected when worms were fed polyP-deficient bacteria (2); (iv) depletion of polyP from plasmatocytes alone impairs hemolymph clotting, which would not be expected if gut-derived polyP were the major source and may have contributed to polyP in hemolymph. We mention this limitation in the revised manuscript (p.19-20, l. 425-433).

(9) Page 3 of Results (paragraph 2): stating that the 4% paraformaldehyde works "best" is imprecise. What do the authors mean by "best"?

We have addressed this comment in the revised manuscript and corrected it as 4% paraformaldehyde being better among the three methods we used to fix tissues, which also included methanol and Bouin’s fixative (p.8, l. 152-154).

(10) Page 4 of Results (paragraph 2, last line of the page): The scientific literature is vast, so one can never be sure that one knows of all the papers out there, even on a topic as relatively limited as polyP. Therefore, I would recommend qualifying the statement "...this is the first comprehensive tissue staining report...". It would be more accurate (and safer) to say something like, "to our knowledge, this is the first..." There is a similar statement with the word "first" on the next page regarding the FLYX library.

We have addressed this concern and corrected it accordingly in the revised version of the manuscript (p.9, l. 192-193)

**Reviewer #3 (Recommendations for the authors):**
(1) The authors should include in their discussion a comparison of cell biological observations using the polyP binding domain of *E. coli* Ppx (GST-PPBD) to fluorescently label polyP in cells and tissues with recent work using a similar approach in *C. elegans* (Quarles et al., PMID:39413779).

In the revised manuscript, we have cited the work of Quarles et al. and have added a comparison of observations (p.19,l.408-410). In the discussion, we have also focused on multiple other studies about how polyP presence in different subcellular compartments, like the nucleus, can be assayed and studied with the tools developed in this study.

(2) The gene expression studies of time-matched Cyto-FLYX vs WT larvae is very intriguing. Given the authors' findings that non-feeding third instar Cyto-FLYX larvae are developmentally ahead of WT larvae, can the observed trends be explained by known changes in gene expression that occur during eclosion? This is mentioned in the results section in the context of genes linked to neurons, but a broader discussion of which pathway changes observed can be explained by the developmental stage difference between the WT and FLYX larvae would be helpful in the discussion.

We have included a broader discussion about the RNAseq analysis done in the article in both the ‘results’ and the ‘discussion’ sections, where we have rewritten the narrative from the perspective of accelerated eclosion. (p.15 l.310-335, p. 20, l.431-446). We have also stated the limitations of such studies.

(3) The sentence describing NUDT3 is not referenced.

We have addressed this comment and have cited the paper of NUDT3 in the revised version of the manuscript.(p.4, l. 68-70)

(4) In the first sentence of the results section, the meaning/validity of the statement "The polyP levels have decreased as evolution progressed" is not clear. It might be more straightforward to give an estimate of the total pmoles polyP/mg protein difference between bacteria/yeast and metazoans.

In the revised manuscript, we have given an estimate of the polyP content across various species across evolution to uphold the statement that polyP levels have decreased as evolution progressed (p. 5, l. 87-91).

(5) The description of the malachite green assay in the results section describes it as "calorimetric" but this should read "colorimetric?"

We have corrected it in the revised manuscript.

References

(1) Chicco D, Agapito G. Nine quick tips for pathway enrichment analysis. PLOS Comput Biol. 2022 Aug 11;18(8):e1010348.

(2) Quarles E, Petreanu L, Narain A, Jain A, Rai A, Wang J, et al. Cryosectioning and immunofluorescence of *C. elegans* reveals endogenous polyphosphate in intestinal endo-lysosomal organelles. Cell Rep Methods. 2024 Oct 8;100879.

(3) Saito K, Ohtomo R, Kuga-Uetake Y, Aono T, Saito M. Direct labeling of polyphosphate at the ultrastructural level in *Saccharomyces cerevisiae* by using the affinity of the polyphosphate binding domain of *Escherichia coli* exopolyphosphatase. Appl Environ Microbiol. 2005 Oct;71(10):5692–701.

(4) Smith SA, Mutch NJ, Baskar D, Rohloff P, Docampo R, Morrissey JH. Polyphosphate modulates blood coagulation and fibrinolysis. Proc Natl Acad Sci USA. 2006 Jan 24;103(4):903–8.

(5) Smith SA, Choi SH, Davis-Harrison R, Huyck J, Boettcher J, Rienstra CM, et al. Polyphosphate exerts differential effects on blood clotting, depending on polymer size. Blood. 2010 Nov 18;116(20):4353–9.

(6) Abramov AY, Fraley C, Diao CT, Winkfein R, Colicos MA, Duchen MR, et al. Targeted polyphosphatase expression alters mitochondrial metabolism and inhibits calcium-dependent cell death. Proc Natl Acad Sci USA. 2007 Nov 13;104(46):18091–6.

(7) Schmid MR, Dziedziech A, Arefin B, Kienzle T, Wang Z, Akhter M, et al. Insect hemolymph coagulation: Kinetics of classically and non-classically secreted clotting factors. Insect Biochem Mol Biol. 2019 Jun;109:63–71.

(8) Jian Guan, Rebecca Lee Hurto, Akash Rai, Christopher A. Azaldegui, Luis A. Ortiz-Rodríguez, Julie S. Biteen, Lydia Freddolino, Ursula Jakob. HP-Bodies – Ancestral Condensates that Regulate RNA Turnover and Protein Translation in Bacteria. bioRxiv 2025.02.06.636932; doi: https://doi.org/10.1101/2025.02.06.636932.

(9) Lonetti A, Szijgyarto Z, Bosch D, Loss O, Azevedo C, Saiardi A. Identification of an evolutionarily conserved family of inorganic polyphosphate endopolyphosphatases. J Biol Chem. 2011 Sep 16;286(37):31966–74.